# AI Stereotypes: An Unequipartition Property for Perplexity in Generative Language Models

## Abstract

We prove a new asymptotic unequipartition property for the perplexity of long texts generated by a language model and present supporting experimental evidence from open-source models. Specifically we show that the logarithmic perplexity of any large text generated by a language model must asymptotically converge to the average entropy of its token distributions. This defines a "typical set" that all long synthetic texts generated by a language model must belong to. We refine the concept of "typical set" to include only grammatically correct texts. We then show that this refined typical set is a vanishingly small subset of all possible grammatically correct texts for a very general definition of grammar. This means that language models are strongly constrained in the range of their possible behaviors and outputs. We make no simplifying assumptions (such as stationarity) about the statistics of language model outputs, and therefore our results are directly applicable to practical real-world models without any approximations. We discuss possible applications of the typical set concept to problems such as detecting synthetic texts and membership inference in training datasets.

## 1 Introduction and Background.

Consider a generative model, defined as an algorithm that takes a user input $\mathbf{X}$ and produces an output $\mathbf{Y}$ that statistically resembles data from a natural source. A specific type of generative model is a large language model (LLM) whose output $\mathbf{Y}$ is a body of text and the input is a text user prompt $\mathbf{X}$. State-of-the-art LLMs (Anthropic, 2024; OpenAI, 2024) are now able to produce detailed and information-rich text outputs such as entire screenplays and book-length manuscripts from a short and simple user prompt. They can also pass the Turing test (Pinar Saygin et al., 2000) i.e. imitate human-written text well enough to be convincing to human observers. In this work, we argue that an LLM is strongly constrained by statistical laws. Specifically, we show that the logarithmic *perplexity* of any large text produced by a language model must asymptotically converge to the average entropy of its token distributions. This means that any language model is constrained to only output text strings from a *typical set*, which we show, is an exponentially vanishing subset of all possible grammatically correct strings. This involves a generalization of the well-known Asymptotic Equipartition Theorem (AEP) (Cover & Thomas, 2005a) from information theory and refining it for language models to account for the concept of grammatical correctness.

This work touches on ideas such as entropy, equipartition theorems and computational linguistics/natural language processing (CL/NLP), each of which boast of a vast body of previous work. We now present a brief literature survey to place our work in context.

### 1.1 Equipartition Theorems

The simplest version of the AEP states that a long sequence of independent and identically distributed (iid) random symbols is likely to be a *typical sequence* (Csiszar, 1998) defined by the property that the empirical distribution of the occurrence of different symbols within the sequence is very close to the distribution from which the symbols were drawn. The AEP itself can be thought of as a consequence of the Law of Large Numbers (LLN) (Idele, 2018) that asserts the convergence of a log-likelihood random variable to a limiting "entropy rate" (Algoet & Cover, 1988).

A weak AEP for iid sequences with a finite alphabet was originally introduced by Shannon (1948) followed by strong versions for stationary, ergodic sequences by McMillan (1953) and Breiman (1957), extensions to infinite alphabets by Chung (1961), and continuous-valued sequences in Barron (1985). Thus, the AEP is also commonly referred to as the Shannon-McMillan-Breiman (SMB) Theorem Hamdan et al. (2008). Generalizations of the SMB Theorem (Moy, 1961) in the form of ergodic theorems are now available for many classes of stationary sequences (Pollicott & Yuri, 1998).

Extending the AEP for iid sequences to a non-stationary sequence of independent, but non-identically distributed symbols is straightforward (Tang et al., 2024). AEPs for word-valued functions of stationary sequences are considered in (Nishiara & Morita, 2000; Timo et al., 2010). However, beyond this, the literature on the AEP for non-stationary sequences is much more limited. Closest to our work is the analysis in Yang & Liu (2004) and Wang & Yang (2016) that consider a very general class of non-homogeneous Markov processes, but assume the existence of a unique limiting distribution for the Markov process to establish a constant entropy rate.

Our key point of departure is to not require equal partitions: without stationarity or similarly restrictive assumptions on the statistics of the process, the likelihoods of long sequences do not all become asymptotically equal. Crucially, we show that the the typical sets defined by the resulting Un-Equipartition Property still retain certain essential properties that make the typical set concept so powerful for stationary processes.

## 1.2 Computational Models of Language

Statistical analysis of languages in its modern form dates back to the work of (Shannon, 1951) on the entropy of written English. These statistical ideas competed (Chomsky, 1959) with structural theories of language (Chomsky, 2014) in the early days of modern linguistics. Early methods such as n-gram models (Manning & Schütze, 1999) evolved into more mathematically sophisticated methods such as Hidden Markov Models (HMMs) (Rabiner, 1989). The statistical approach has now become dominant after the recent introduction of neural network architectures (Bengio et al., 2003), culminating in the development of powerful large language models (LLMs) based on transformer networks (Vaswani et al., 2017). While we know that LLMs can generate complex and coherent text outputs, their representation of language structures and grammar remains a topic of research Reizinger et al. (2024); Mészáros et al. (2024).

Perplexity, defined as an inverse likelihood function, is widely used as a performance metric for training language models (Meister & Cotterell, 2021). It is closely related to the information-theoretic concepts of *surprisal* (Levy, 2008) and entropy (Cover & Thomas, 2005b), and it also appears to capture linguistic (Miaschi et al., 2021; Gamallo et al., 2017) and cognitive (Demberg & Keller, 2008; Cohen & Pakhomov, 2020) phenomena at least partially. Many "AI detection" tools are based on observed differences between the perplexity of synthetic and natural text (Mitchell et al., 2023; Gehrmann et al., 2019).

## 1.3 Contribution: An Un-Equipartition Property for Unnatural Language

Our main contributions are (a) to formulate and provide theoretical and experimental justification for a version of the AEP that applies to real-world practical LLMs without any approximations, and (b) use this AEP to define a concept of typical set specifically adapted for language models. While our focus is on the theoretical development, we also briefly discuss possible applications to practical problems such as AI text detection and LLM dataset inference.

Unlike computational linguistics, we do not study natural language; instead our exclusive focus is on *unnatural language* i.e. synthetic text created by LMMs. Natural languages are shaped by complex social, psychological and biological processes Pierce (1968) and they can only ever be approximated by computational models Bar-Hillel (1962). In contrast, LLMs are machines whose internals, while complex, are in theory completely knowable.

Thus, it is possible for us to derive *exact* laws that govern the outputs of LLMs in a way that is not possible to do for natural language. However, this requires us to be disciplined about not making any assumptions or approximations about the statistics of LLM outputs.

### 1.4 Notation

A sequence of symbols is denoted by boldface, uppercase identifiers e.g. $\mathbf{A}$, $\mathbf{B}$. These can be fixed, constant sequences or random sequences. The vector consisting of the first $N$ elements of a sequence $\mathbf{A}$ is denoted by $\mathbf{A}_N \equiv [A_1, A_2, \ldots, A_N]$. All random variables in this work take discrete-values from a finite set. The probability of an event $E$ is $\Pr(E)$, the distribution or probability mass function of a random variable $A_m$ is $p(A_m)$. Sets are denoted by calligraphic symbols e.g. $\mathcal{A}$. The binary entropy of a distribution $p(A_m)$ is $H(p(A_m))$[1]. The symbol $\equiv$ denotes a simple identity while $=$ denotes a mathematical inference and $\doteq$ a definition.

## 2 Definitions and Problem Statement

Let M be a generative model whose output $\mathbf{Y}$ consisting of a sequence of tokens $\mathbf{Y} = [Y_1, Y_2, \ldots Y_N \ldots]$, chosen from a finite token dictionary $Y_n \in \mathcal{Y}$, is a stochastic function of user prompt $\mathbf{X}$. The model M is defined by a probability distribution $p(\mathbf{Y}|\mathbf{X})$, or more formally, a sequence of probability distributions $p(\mathbf{Y}_N|\mathbf{X})$ over $\mathbf{Y}_N \in \mathcal{Y}^N$ where $\mathbf{Y}_N \doteq [Y_1, Y_2, \ldots Y_N]$ is the substring of $\mathbf{Y}$ consisting of the first $N$ tokens. Practical implementations of LLMs specify the probability distribution iteratively (Radford et al., 2018):

$$p(Y_1, Y_2|\mathbf{X}) = p(Y_1|\mathbf{X})p(Y_2|Y_1, \mathbf{X}) \tag{1}$$

and so on. Thus, the model M first draws a random value for the first token say $Y_1 = y_1$ by sampling from the distribution $p(Y_1|\mathbf{X})$. Then the model determines a distribution for the second token as a function of the initial prompt $\mathbf{X}$ and the randomly chosen first token $y_1$. Thus, the second token is randomly sampled from a distribution $p(Y_2|\mathbf{X}, Y_1 = y_1)$ and so on. We can write

$$p(\mathbf{Y}_N|\mathbf{X}) = p_1(Y_1)p_2(Y_2)\ldots p_N(Y_N), \text{ where } p_n(Y_n) = p(Y_n|Y_1, Y_2, Y_{n-1}, \mathbf{X}) \tag{2}$$

Given a string $\mathbf{Y}$, open-source LLMs can be programmed to print out the distributions $p_n(Y_n)$ from which its tokens were selected. Specifically, given a user prompt $\mathbf{X}$ and a string of tokens $\mathbf{Y}_N \equiv [Y_1, Y_2, \ldots, Y_N]$, it is possible to get a complete listing of the distributions $p_n(Y_n)$, $n = 1 \ldots N$.

**Remark.** Equation (2) is simply an application of the Bayes rule of probability theory and it always holds for any generative model regardless of whether the tokens $Y_n$ are sequentially generated. However, the conditional distributions $p_n(y)$ are not in general easily accessible, so while (2) is true for all generative models, it may only be *useful* for sequential models.

The perplexity $\text{perp}_M(\mathbf{Y}_N) \doteq \prod_{n=1}^N p_n(Y_n)^{-\frac{1}{N}}$ of a text string $\mathbf{Y}_N = [Y_1, Y_2, \ldots Y_N]$ for a model M is defined as the per-token inverse likelihood of the string $\mathbf{Y}$. It is usually more convenient to work with the log-perplexity $l_M(\mathbf{Y}_N)$:

$$l_M(\mathbf{Y}_N) \doteq \log_2(\text{perp}_M(\mathbf{Y}_N)) \equiv -\frac{1}{N}\sum_{n=1}^N \log_2(p_n(Y_n)) \tag{3}$$

### 2.1 A Toy Problem

Let $\alpha(y)$, $\beta(y)$ be two fixed probability distributions over the (discrete) set $\mathcal{Y}$ of tokens. Consider a toy problem involving two language models A and B, that each generate a string of tokens $\mathbf{Y} = [Y_1, Y_2, \ldots]$ where each token is generated iid from the distribution $\alpha(y)$ and $\beta(y)$ respectively. The iid assumption implies that the tokens $Y_n$ can be thought of as being generated by a stationary and ergodic random process[2].

Consider a long string $\mathbf{A}_N \doteq [A_1, A_2, \ldots, A_N]$, $N \gg 1$ randomly generated from model A. Let $p_{\mathbf{A}}(y)$ denote the empirical distribution of $y \in \mathcal{Y}$ in the string $\mathbf{A}_N$:

$$p_{\mathbf{A}}(y) \doteq \frac{n_{\mathbf{A}}(y)}{N} \equiv \frac{1}{N}\sum_{n=1}^N \mathbf{1}(A_n \equiv y), \; \forall y \in \mathcal{Y} \tag{4}$$

---

[1]We will always speak of the entropy of distributions and avoid speaking of the entropy of random variables.
[2]Strictly speaking, a stationary random process "starts" at $n = -\infty$ rather than $n = 1$; here we ignore the tokens produced by the assumed random process before $n = 1$.

where $\mathbf{1}(.)$ denotes the indicator function and $n_{\mathbf{A}}(y)$ is the number of occurrences of token $y$ in the (long) string $\mathbf{A}_N$. The log-perplexity of string $\mathbf{A}_N$ for model A is:

$$l_A(\mathbf{A}_N) = -\frac{1}{N}\sum_{n=1}^{N}\log_2(\alpha(Y_n)) \equiv -\sum_{y\in\mathcal{Y}}\frac{n_{\mathbf{A}}(y)}{N}\log_2(\alpha(y)) \equiv H(p_{\mathbf{A}},\alpha) \tag{5}$$

where $H(\gamma_1,\gamma_2) \doteq -\sum_{y\in\mathcal{Y}}\gamma_1(y)\log_2(\gamma_2(y))$ is the cross-entropy between distributions $\gamma_1(y)$, $\gamma_2(y)$ over $y \in \mathcal{Y}$. It is well-known $H(\gamma_1,\gamma_2) \geq H(\gamma_1)$ with equality when $\gamma_1 \equiv \gamma_2$, where $H(\gamma) \doteq -\sum_{y\in\mathcal{Y}}\gamma(y)\log_2(\gamma(y))$ is the entropy of distribution $\gamma$. Thus we have from (5):

$$l_A(\mathbf{A}_N) \equiv H(p_{\mathbf{A}},\alpha) \geq H(p_{\mathbf{A}})\text{with equality iff } p_{\mathbf{A}}(y) \equiv \alpha(y),\ \forall y \in \mathcal{Y} \tag{6}$$

The simplest version of the classical Asymptotic Equipartition Theorem (AEP) (Cover & Thomas, 2005a) from information theory states that the log-perplexity of a long string $\mathbf{A}_N$ of iid symbols is almost always very close to the entropy $H(\alpha)$ of the distribution $\alpha(y)$ the symbols are drawn from.

**Proposition 2.1. A Simple Cross-AEP.** *For a long text string $\mathbf{A}_N = [A_1,\ A_2,\ \ldots,\ A_N]$ of iid tokens $A_n$ drawn from a distribution $\alpha(y)$, the log perplexity $l_B(\mathbf{A}_N)$ for a model B as defined in (3) almost always approaches the cross-entropy of the distributions $H(\alpha,\beta)$:*

$$\lim_{N\to\infty} l_B(\mathbf{A}_N) \equiv H(\alpha,\beta),\ \text{and}\ \lim_{N\to\infty} l_A(\mathbf{A}_N) \equiv H(\alpha) \tag{7}$$

From (5) and (6), we see that (7) itself is equivalent to $p_{\mathbf{A}}(y) \equiv \frac{n_{\mathbf{A}}(y)}{N} \to \alpha(y),\ \forall y \in \mathcal{Y}$ i.e. the empirical distribution $p_{\mathbf{A}}(y)$ for long strings $\mathbf{A}$ almost always converges to $\alpha(y)$. Putting these observations together we have:

$$\lim_{N\to\infty} l_A(\mathbf{A}_N) \equiv H(\alpha) \leq H(\alpha,\beta) \equiv \lim_{N\to\infty} l_B(\mathbf{A}_N) \tag{8}$$

Intuitively, (8) states that a long string $\mathbf{A}_N$ generated from model A is likely to have a lower perplexity for model A than for any other model B. This means that a model trained to have low perplexity over a set of reference texts will generate text strings that are statistically similar to the reference texts. This is the theoretical justification for using perplexity as a loss function for training language models and it is an excellent justification - provided only that we accept the Shannon model of language i.e. the idea that languages can be reasonably modeled as a stochastic sequence of tokens (Jakobson, 1961). Testing for abnormally high or low perplexity is an important tool for LLM applications as we discuss in Section 6.

## 2.2 A Modest Generalization

The simple AEP in Proposition 2.1 has been extended in various ways in the literature such as the following version (see Tal et al. (2017), Appendix B.4), which we will make use of in the sequel.

**Proposition 2.2. Generalized AEP.** *Consider a language model A' that generates text string $\mathbf{A}'_N = [A'_1,\ A'_2,\ \ldots,\ A'_N]$ where the tokens $A'_n$ are drawn independently from a sequence of distributions $\alpha_n(y)$, $n = 1, 2, \ldots$. Assuming the entropies of distributions $\alpha_n(y)$ are uniformly bounded i.e. $H(\alpha_n(y)) < H_{max} < \infty,\ \forall n$, the log perplexity $l_{A'}(\mathbf{A}'_N)$ of the string $\mathbf{A}'_N$ as defined in (3) is close to the average entropy of the distributions $\alpha_n(y)$, $n = 1, 2, \ldots, N$ with high probability:*

$$\lim_{N\to\infty}\Pr[|l_{A'}(\mathbf{A}'_N) - h_{A'}(N)| > \epsilon] \equiv 0,\ \forall\epsilon > 0, \text{where } h_{A'}(N) \equiv \frac{1}{N}\sum_{n=1}^{N}H(\alpha_n) \tag{9}$$

The regularity condition that the entropies of $\alpha_n(y)$ are uniformly bounded is trivially true if the token dictionary $\mathcal{Y}$ is a finite set: $H(p(y)) \leq \log_2|\mathcal{Y}|$ for any distribution $p(y)$ over $y \in \mathcal{Y}$.

While Proposition 2.2 does not lend itself to an intuitive interpretation in terms of the empirical distribution of the tokens $A'_n$, it too is a direct consequence of the Law of Large Numbers applied to the log-perplexity

random variable. Much of the analysis in Section 2.1 can be extended to generalized models like $A'$ in Proposition 2.2 that allows each token $A'_n$ to be drawn from different distributions $\alpha_n(y)$. However, the tokens in these models must be drawn from *fixed distributions independent of past tokens*. This is still quite trivial compared to modern language models where the output tokens depend in highly complex and sophisticated ways on past tokens.

## 3 An Un-Equipartition Property for Perplexity

We now propose a generalization of the theory described in Section 2.1 to a more interesting class of models. Consider a language model M and a random infinitely long text string $\mathbf{Y}$ generated by M, whose probabilities for a given prompt $\mathbf{X}$ is described by (2). Specifically, given a model M and a string $\mathbf{Y}$, (2) defines a sequence of probability distributions $p_n(y)$ over the set of tokens $y \in \mathcal{Y}$. We will assume that the prompt $\mathbf{X}$ is fixed and omit it from our notation in the sequel.

**Definition 3.1.** The empirical entropy $h_M(\mathbf{Y}_N)$ of model M for string $\mathbf{Y}_N$ is defined as:

$$h_M(\mathbf{Y}_N) \doteq \frac{1}{N} \sum_{n=1}^{N} H(p_n) \tag{10}$$

where $H(p) \equiv -\sum_{y \in \mathcal{Y}} p(y) \log_2(p(y))$ is the entropy of the distribution $p(y)$ over $y \in \mathcal{Y}$.

Note that we make no assumptions about the existence of an "entropy rate" $\lim_{N \to \infty} h_M(\mathbf{Y}_N)$ for the model $M$. However, we can show that

$$E[l_M(\mathbf{Y}_N)] \equiv E[h_M(\mathbf{Y}_N)] \equiv \frac{1}{N} H(\Pr(\mathbf{Y}_N)) \tag{11}$$

where $H(\Pr(\mathbf{Y}_N))$ is the entropy of the random text strings $\mathbf{Y}_N$.

**Definition 3.2.** Let the log-deviation $\lambda(p)$ of a probability distribution $p(y)$ over $y \in \mathcal{Y}$ be defined as:

$$\lambda(p) \doteq \sqrt{S(p) - (H(p))^2}, \text{ where } S(p) \doteq \sum_{y \in \mathcal{Y}} p(y) \left(\log_2 p(y)\right)^2 \tag{12}$$

Note that the entropy $H(p)$ and log-deviation $\lambda(p)$ are the mean and standard deviation of the log-likelihood random variable $l_p(y) \equiv -\log_2 p(y)$ under distribution $p(y)$.

**Definition 3.3.** The log-deviation $\lambda_M(\mathbf{Y}_N)$ of a string $\mathbf{Y}_N$ for model $M$ is defined as:

$$\lambda_M(\mathbf{Y}_N) \doteq \frac{1}{N} \sqrt{\sum_{n=1}^{N} \lambda^2(p_n)} \tag{13}$$

We can again interpret $H(p_n)$, $\lambda(p_n)$ as the *conditional* mean and standard deviation of $-\log_2(\Pr(Y_n))$ conditioned on $\mathbf{Y}_{N-1}$. Clearly, if the log-deviations $\lambda(p_n)$ of the distributions $p_n(y)$ for a string $\mathbf{Y}$ are uniformly upper-bounded, $\lambda_M(\mathbf{Y}_N)$ asymptotically vanishes. We are now ready to state our main result.

**Proposition 3.1. Un-Equipartition Property for Perplexity.** *For a long text string $\mathbf{Y}$ i.e. $N \gg 1$ generated from a language model with a given prompt $\mathbf{X}$, with high probability, the log perplexity $l_M(\mathbf{Y})$ is close to the empirical entropy $h_M(\mathbf{Y})$. More precisely:*

$$\lim_{N \to \infty} \Pr[|l_M(\mathbf{Y}_N) - h_M(\mathbf{Y}_N)| > \epsilon] \equiv 0, \ \forall \epsilon > 0 \tag{14}$$

We want to argue that $\lambda_M(\mathbf{Y}_N)$ is the standard deviation of $l_M(\mathbf{Y}_N) - h_M(\mathbf{Y}_N)$, and therefore asymptotically vanishing $\lambda_M(\mathbf{Y}_N)$ implies $l_M(\mathbf{Y}_N) \to h_M(\mathbf{Y}_N)$. This, however, is very wrong - for one thing $\lambda_M(\mathbf{Y}_N)$ is itself a random variable. In the sequel, we show an informal argument to carry through reasoning like the above. But first we present a direct proof of Proposition 3.1.

### 3.1 Formal Proof: an LLN for LLMs

We will use a version of the Weak Law of Large Numbers that applies to sequences of random variables with no restrictions on their dependence. Such a Law can be obtained using martingale difference ideas and an elementary derivation is presented in Appendix A.

*Proof of Proposition 3.1.* Consider the sequence of conditional likelihood random variables $\{l_n\}$ defined as:

$$l_n \doteq -\log_2 \Pr\left(Y_n | \mathbf{Y}_{n-1}\right) \equiv -\log_2\left(p_n(Y_n)\right) \tag{15}$$

We note that the random variables $Z_n \doteq l_n$ satisfy all of the conditions for Lemma A.2, if we impose the restriction that they are strictly bounded i.e. $|l_n| < L$, for some some finite $L$. This is equivalent to requiring that all non-zero token probabilities $p_n(y)$ are above some lower bound e.g. $p_n(y) \geq 10^{-10}$. In practice, this is easily satisfied by any practical LLM. Then from Lemma A.2, we have for any $\epsilon > 0$:

$$\lim_{N \to \infty} \Pr\left(\left|\frac{1}{N}\sum_{n=1}^{N}(l_n - H(p_n))\right| > \epsilon\right) = 0 \tag{16}$$

From the definition (15), we note that $\frac{1}{N}\sum_{n=1}^{N} l_n \equiv l_M(\mathbf{Y}_N)$, and likewise $\frac{1}{N}\sum_{n=1}^{N} H(p_n) \equiv h_M(\mathbf{Y}_N)$. Combining these observations with (16) gives (14). $\qquad\square$

### 3.2 Informal Proof: Roads not Traveled

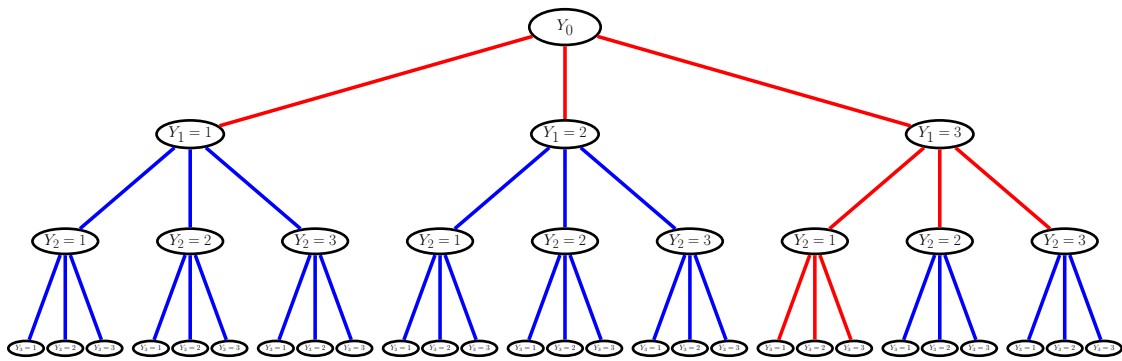

Figure 1: Probability tree for the first 3 tokens of a language model M that selects each token $Y_n$ from a dictionary $\mathcal{Y}$ of three tokens. Also shown in red is the probability tree of an auxiliary model $M'_{\mathbf{S}}$ for a string $\mathbf{S}$ with $S_1 = 3$, $S_2 = 1$ and so on.

Consider a fixed string $\mathbf{S}$. For this particular string $\mathbf{S}$ and model M, we have a fixed set of distributions $q_n(y) \doteq \Pr(S_n = y | \mathbf{S}_{n-1})$. Now define an auxiliary generative model $M'_{\mathbf{S}}$ that generates a random string $\mathbf{Y}' \doteq [Y'_1, Y'_2, \ldots, Y'_N, \ldots]$ where the tokens $Y'_n$ are generated *independently* from the distributions $q_n(y)$. If we represent a language model M as a probability tree, an auxiliary model $M'_{\mathbf{S}}$ represents one specific subtree corresponding to the string $\mathbf{S}$ that defines the model. This is illustrated in Fig. 1. The auxiliary model $M'_{\mathbf{S}}$ is much simpler than the general language model M, because its output tokens $Y'_n$ are generated independently of other tokens. Our informal proof of Proposition 3.1 is based on the following idea: while calculating $\Pr(E)$ for some event $E$, we can change any part of the probability tree that do not affect the event $E$ ("the roads not traveled") to make our calculation easier e.g. by allowing us to work with an auxiliary language model $M'_{\mathbf{S}}$ which generates strings of independent tokens.

*Informal Proof of Proposition 3.1.* Assuming the tokens $Y_n$ are generated by the auxiliary model $M'_{\mathbf{S}}$ (instead of the original model M), the random variables $l_m$, $l_n$ are independent and $\mathrm{E}[l_n] \equiv H(q_n)$. Equation (16) then immediately follows from Proposition 2.2. $\qquad\square$

This informal method allows us to treat the martingale property that the formal proof relies on as a quasi-independence condition to reason about the random variables $l_n$ intuitively. As an example, for the auxiliary model $M'_{\mathbf{S}}$, from (12, 13), the variances of $l_n$, $l_M(\mathbf{S}_N)$ are respectively $\lambda^2(q_n)$, $\lambda_M^2(\mathbf{S}_N)$. We then have the following simple bound on deviations using the Chebyshev's Inequality Cohen (2015):

$$\Pr\left(|l_M(\mathbf{S}_N) - h_M(\mathbf{S}_N)| > \alpha\lambda_M(\mathbf{S}_N)\right) \leq \frac{1}{\alpha^2}, \ \forall \alpha > 0 \tag{17}$$

### 3.3 Typical Sets

We can now formally define the *typical set* $\mathcal{T}_M^n(\epsilon) \subset \mathcal{Y}^n$ of *typical strings* $\mathbf{Y}_n$ for a model M with $\epsilon > 0$:

$$\mathcal{T}_M^n(\epsilon) \doteq \left\{\mathbf{Y}_n : |l_M(\mathbf{Y}_n) - h_M(\mathbf{Y}_n)| < \epsilon\right\} \equiv \left\{\mathbf{Y}_n : \left|h_M(\mathbf{Y}_n) + \frac{1}{n}\log_2\left(\Pr(\mathbf{Y}_n)\right)\right| < \epsilon\right\} \tag{18}$$

Proposition 3.1 asserts that with high probability a stochastic model $M$ is constrained to only output typical strings from this set $\mathcal{T}_M^n(\epsilon)$ for any $\epsilon > 0$ if we consider sufficiently long strings. In practical applications of the typical set concept, we almost invariably find that the typical set is *vanishingly small*. This is what makes the typical set concept so powerful: it identifies the small number of outcomes that actually matter, so we can focus our resources on them efficiently.

We will show that here too, the typical set must be vanishingly small under mild assumptions. Since the vast majority of token strings represent gibberish text, it is trivially true that any LLM that generates coherent text will not generate most token strings. Thus, the simple notion of smallness - that the typical set is a vanishing subset of all token strings - is not very interesting.

We want to show a much stronger result that even if we exclude gibberish sentences and only look at the set of *grammatically correct* texts, any practical LLM can only generate a vanishingly small fraction of all possible grammatical texts. This requires additional analytical machinery which we now introduce.

## 4 The Grammar Police

Let $\mathcal{G}(n)$ denote a dictionary of "grammatical" strings $\mathbf{Y}_n$ of length $n$ and $g(n) \doteq \frac{1}{n}\log_2|\mathcal{G}(n)|$. Since most token strings represent meaningless gibberish, we expect $|\mathcal{G}(n)| \ll |\mathcal{Y}|^n$ or $g(n) \ll \log_2|\mathcal{Y}|$ for any reasonable notion of grammar. However, we also want our dictionary to be rich enough to include sentences of unlimited length and complexity to avoid trivialities. Formally we require that $g(n) \geq g_{min} > 0$. Beyond this, our analysis below is very general, which means we can define the notion of "grammatical strings" to be essentially anything we want.

### 4.1 Augmented Typical Sets for Language Models

We can now define the augmented typical set of the model $M$ for dictionary $\mathcal{G}(n)$ as:

$$\mathcal{T}_{M,G}^n(\epsilon) \doteq \mathcal{G}(n) \cap \mathcal{T}_M^n(\epsilon) \equiv \left\{\mathbf{Y}_n \in \mathcal{G}(n) : |l_M(\mathbf{Y}_n) - h_M(\mathbf{Y}_n)| < \epsilon\right\} \tag{19}$$

Let $p_G(n) \doteq \Pr(\mathbf{Y}_n \in \mathcal{G}(n))$ be the probability that model $M$ generates a grammatically correct string $\mathbf{Y}_n \in \mathcal{G}(n)$. We will assume that $p_G(n) \geq p_G > 0$ i.e. the model generates grammatical strings of any length with a non-zero probability. Informally, the model $M$ knows how to generate grammatically correct sentences when suitably prompted, but we do not require it to be perfect and it may still sometimes generate ungrammatical outputs. The probability that the model $M$ generates a grammatical string $\mathbf{Y}_n \in \mathcal{G}(n)$ can be written as:

$$\Pr\left(\mathbf{Y}_n \middle| \mathbf{Y}_n \in \mathcal{G}(n)\right) = \frac{\Pr\left(\mathbf{Y}_n\right)}{p_G(n)} \equiv \frac{2^{-nl_M(\mathbf{Y}_n)}}{p_G(n)} \tag{20}$$

We can use (20) to relate the perplexity of grammatical strings specifically to the entropy analogous to (11):

$$\frac{1}{n}H\left(\Pr(\mathbf{Y}_n)\middle|\mathbf{Y}_n \in \mathcal{G}(n)\right) \equiv \frac{\log_2 p_G(n)}{n} + E\left[l_M(\mathbf{Y}_n)\middle|\mathbf{Y}_n \in \mathcal{G}(n)\right] \leq g(n) \tag{21}$$

where the last inequality holds with equality for a model that selects all grammatical texts with equal probability: $\Pr(\mathbf{Y}_n) \equiv \frac{1}{|\mathcal{G}(n)|}$, $\forall \mathbf{Y}_n \in \mathcal{G}(n)$ (for which $p_G(n) = 1$). Unlike (11), this does not tell us anything about the empirical entropy $h_M(\mathbf{Y}_n)$ of individual strings $\mathbf{Y}_n$. We will now show that we can safely ignore high entropy strings.

**Proposition 4.1. Entropies cannot be too large.** *The empirical entropy $h_M(\mathbf{Y}_n)$ of any long grammatical string $\mathbf{Y}_n \in \mathcal{G}(n)$ generated by the model $M$ is greater than $g(n) \equiv \frac{1}{n} \log_2 |\mathcal{G}(n)|$ with vanishing probability:*

$$\lim_{n \to \infty} \Pr\left(h_M(\mathbf{Y}_n) > g(n) + \epsilon \big| \mathbf{Y}_n \in \mathcal{G}(n)\right) = 0, \ \forall \epsilon > 0 \tag{22}$$

*Proof.* First we note that for sufficiently large $n$, we must have with high probability $|l_M(\mathbf{Y}_n) - h_M(\mathbf{Y}_n)| < \frac{\epsilon}{2}$. Then we have:

$$\Pr\left(h_M(\mathbf{Y}_n) > g(n) + \epsilon \big| \mathbf{Y}_n \in \mathcal{G}(n)\right) \leq \frac{1}{p_G(n)} \Pr\left(\mathcal{E}_1(n)\right) \leq \frac{1}{p_G} \Pr\left(\mathcal{E}_1(n)\right)$$

where $\mathcal{E}_1(n) \doteq \left\{\mathbf{Y}_n \in \mathcal{G}(n) : l_M(\mathbf{Y}_n) > g(n) + \frac{\epsilon}{2}\right\}$. We can then bound this probability as:

$$\Pr(\mathcal{E}_1(n)) = \sum_{\mathbf{Y}_n \in \mathcal{E}_1(n)} \Pr(\mathbf{Y}_n) \leq \sum_{\mathbf{Y}_n \in \mathcal{E}_1(n)} 2^{-n(g(n)+\frac{\epsilon}{2})} \equiv |\mathcal{E}_1(n)| \, 2^{-n(g(n)+\frac{\epsilon}{2})} < |\mathcal{G}(n)| \, 2^{-n(g(n)+\frac{\epsilon}{2})} \equiv 2^{-\frac{n\epsilon}{2}}$$

Thus we have shown that $\Pr\left(h_M(\mathbf{Y}_n) > g(n) + \epsilon \big| \mathbf{Y}_n \in \mathcal{G}(n)\right) < \frac{1}{p_G} 2^{-\frac{n\epsilon}{2}} \to 0$ as $n \to \infty$. $\qquad \square$

Proposition 4.1 allows us to purge high entropy strings out of the typical set while retaining all of its probability mass. Specifically, we define the purged typical set $\mathcal{P}_{M,G}^n(\epsilon)$ as:

$$\mathcal{P}_{M,G}^n(\epsilon) \doteq \mathcal{G}(n) \cap \mathcal{T}_M^n(\epsilon) \cap \mathcal{E}_1^c(n), \text{ where } \lim_{n \to \infty} \Pr\left(\mathcal{P}_{M,G}^n(\epsilon) | \mathbf{Y}_n \in \mathcal{G}(n)\right) = 1, \ \forall \epsilon > 0 \tag{23}$$

## 4.2 The Essential Smallness of Typical Sets

For an entropy-maximizing model that chooses all grammatical strings with equal probability, the typical set includes all grammatical strings $\mathcal{P}_{M,G}^n(\epsilon) \equiv \mathcal{G}(n)$. We now formalize the assumption that practical LLMs do not do this, and for all such models, we will show that the typical set is a vanishingly small subset of $\mathcal{G}(n)$.

**Assumption 4.1. Practical LLMs are not entropy-maximizing.** The empirical entropy $h_M(\mathbf{Y}_n)$ is strictly smaller than $g(n)$ for at least some fraction of grammatical typical strings. Specifically, there is an entropy gap $\Delta g > 0$ such that $h_M(\mathbf{Y}_n) \leq g(n) - \Delta g$ for at least some non-zero fraction $\rho > 0$ of typical strings $\mathbf{Y}_n$:

$$|\mathcal{E}_2(n)| \geq \rho \left|\mathcal{P}_{M,G}^n(\epsilon)\right|, \text{ where } \mathcal{E}_2(n) \doteq \left\{\mathbf{Y}_n \in \mathcal{P}_{M,G}^n(\epsilon) : h_M(\mathbf{Y}_n) \leq g(n) - \Delta g\right\} \tag{24}$$

**Proposition 4.2. Typical sets are small.** *Under Assumption 4.1, the size of the pruned typical set $\mathcal{P}_{M,G}^n$ is an exponentially vanishing fraction of all grammatically correct sequences $\mathcal{G}(n)$.*

*Proof.* From Proposition 3.1, for sufficiently large $n$, we have $|l_M(\mathbf{Y}_n) - h_M(\mathbf{Y}_n)| \leq \frac{\Delta g}{2}$. Using this observation for $\mathbf{Y}_n \in \mathcal{E}_2(n)$, we have $l_M(\mathbf{Y}_n) \leq g(n) - \frac{\Delta g}{2}$. Then we have:

$$1 \geq \Pr(\mathcal{E}_2(n)) \equiv \sum_{\mathbf{Y}_n \in \mathcal{E}_2(n)} 2^{-n l_M(\mathbf{Y}_n)} \geq \sum_{\mathbf{Y}_n \in \mathcal{E}_2(n)} 2^{-n\left(g(n)-\frac{\Delta g}{2}\right)} \equiv \frac{|\mathcal{E}_2(n)|}{|\mathcal{G}(n)|} 2^{\frac{n\Delta g}{2}} \geq \frac{\rho |\mathcal{P}_{M,G}^n(\epsilon)|}{|\mathcal{G}(n)|} 2^{\frac{n\Delta g}{2}} \tag{25}$$

Thus, we have shown $|\mathcal{P}_{M,G}^n(\epsilon)| \leq |\mathcal{G}(n)| 2^{-n\left(\frac{\Delta g}{2} - \frac{\log_2(\rho)}{n}\right)}$. $\qquad \square$

The obverse of Proposition 4.2 is that almost all grammatically correct strings are non-typical for a given language model M. A string can be non-typical in two different ways depending on whether its perplexity is abnormally high ("under-typical") or abnormally low ("over-typical"). Formally, we define the set $\mathcal{V}_M^n(\epsilon)$ of over-typical strings of length $n$ for a model M as:

$$\mathcal{V}_M^n(\epsilon) \doteq \left\{\mathbf{Y}_n \in \mathcal{G}(n) : l_M(\mathbf{Y}_n) \leq h_M(\mathbf{Y}_n) - \epsilon < g(n) - \epsilon\right\} \tag{26}$$

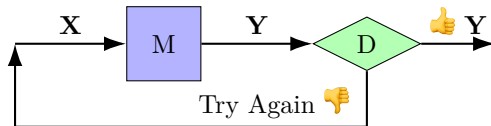

Figure 2: Language Model Augmented with a Grammar Checking Device.

**Proposition 4.3. Over-Typical sets are smaller still.** *The size of the over-typical set $\mathcal{V}_M^n(\epsilon)$ is an exponentially vanishing fraction of all grammatically correct sequences.*

*Proof.* The argument is similar to the proof of Proposition 4.2:

$$1 \geq \Pr\left(\mathbf{Y}_n \in \mathcal{V}_M^n(\epsilon)\right) \equiv \sum_{\mathbf{Y}_n \in \mathcal{V}_M^n(\epsilon)} 2^{-n l_M(\mathbf{Y}_n)} \geq \left|\mathcal{V}_M^n(\epsilon)\right| 2^{-n(g(n)-\epsilon)} \equiv \frac{\left|\mathcal{V}_M^n(\epsilon)\right|}{\left|\mathcal{G}(n)\right|} 2^{n\epsilon} \tag{27}$$

Thus we have $\left|\mathcal{V}_M^n(\epsilon)\right| \leq \left|\mathcal{G}(n)\right| 2^{-n\epsilon}$ which proves the result. $\qquad\square$

For Assumption 4.1 to be violated, we need the perplexities of almost all typical strings to be almost equal; conversely, any model that chooses some strings with higher probabilities than others must satisfy Assumption 4.1 and therefore Proposition 4.2. Together with Proposition 4.3, this means that almost all grammatical strings are *undertypical* for any practical LLM.

### 4.3 Constriction by Staying True to Type

We now present some example consequences of Proposition 4.2 as a payoff for the preceding analysis. Informally, Proposition 3.1 says that all models must stay "true to type", and Proposition 4.2 says that this narrowly circumscribes the range of a model's possible behaviors and outputs.

Consider the augmented language model $G = (M, D)$ shown in Fig. 2, where the output of a model $M$ is screened by a grammar-checking device $D$ which defines the dictionary $\mathcal{G}(n)$; if the model's output $\mathbf{Y}$ is not grammatically correct, then it is prompted again to produce a new random string $\mathbf{Y}$ until it generates a grammatical string[3]. The construction in Fig. 2 implicitly assumes a *decidable* grammar, which means it describes a "recursive language" in Type 1 of the Chomsky hierarchy - the same category natural languages are believed to belong to (Chomsky, 2014).

Since $p_G(n) > p_G > 0$, the augmented model $G$ is guaranteed to generate a grammatical string in a finite time with probability one and the probability that model $G$ generates a grammatical string $\mathbf{Y}_n$ is $\Pr(\mathbf{Y}_n | \mathbf{Y}_n \in \mathcal{G}(n))$. Furthermore, we see from (23) that the augmented model $G$ will only generate outputs from the set $\mathcal{P}_{M,G}^n(\epsilon)$ with high probability. We can then interpret the Proposition 4.2 as a restriction on what the augmented model $G$ can generate as illustrated in the following examples:

Ex 1 **The Python programming language.** We define $\mathcal{G}(n)$ as the set of all syntactically correct Python code that compiles (or interprets) without error. While we know LLMs are capable of generating complex Python code, Proposition 4.2 asserts that any model is limited to generating a vanishing fraction of all valid Python programs.

Ex 2 **Proofs in First Order Logic (FOL).** We define $\mathcal{G}(n)$ as the set of valid mathematical proofs in FOL. A model used to generate proofs in FOL is limited to generating a vanishing fraction of all correct proofs.

Ex 3 **Lyrical Poetry.** We define $\mathcal{G}(n)$ as the set of stories consisting of a sequence of haikus. A model is limited to a vanishing fraction of all such possible stories.

---

[3]Note that we assume that the device $D$ is deterministic, which means our analysis does not allow the grammar checking device to be another stochastic LLM.

# 5 Experiments with Open-Source Models

We performed a series of experiments to verify the ideas described in Sections 3 and 4. The basic idea is to use a local instance of a pre-trained open-source model to evaluate the empirical entropy of a given string using the model's conditional probability distributions and compare the result with the log-perplexity of the string again calculated using the model's probability distributions. We used the smaller GPT-2 model (Radford et al., 2019) as well as the larger and more capable Llama 3.1 8B (Touvron et al., 2023) model.

## 5.1 Perplexity of Self-Generated Strings

Our first set of experiments consisted of repeatedly generating a long text string from a model (GPT-2 or Llama 3.1 8B) and then evaluating its log-perplexity, empirical entropy and log-deviations for the same model.

We initialized the prompt string and the seed of random number generator with fixed values to allow for later reproduction. We then used top-k sampling with $k = 100$ to generate the probability distribution for the next token. We normalized the probabilities of the top-k tokens to sum to unity. We then created and saved a table consisting of token ID, token string and selection probability of each of the top $k$ tokens. Finally, a random next token is chosen by sampling from the normalized distribution, and the new token is appended to the prompt to repeat the process for $N_{max}$ total tokens. We want the number of tokens $N_{max}$ to be large enough to observe the asymptotic behavior of the perplexity of the generated string.

The random token $Y_n$ chosen at each step along with the probability distribution $p_n(y)$ from which they were chosen were saved into a json file for later processing. We ran this experiment many times for different seed values for the random number generator, different choices of sampling distributions and initial prompts.

For each generated string, we can then test our claim in Proposition 3.1. Specifically, for each token $N \in 1 \ldots N_{max}$ we calculated the log-perplexity $l_M(\mathbf{Y}_N)$, empirical entropy $h_M(\mathbf{Y}_N)$ and the log-deviation $\lambda_M(\mathbf{Y}_N)$ of the substring $\mathbf{Y}_N$ consisting of the first $N$ tokens of the generated string.

## 5.2 Perplexity of Externally Generated Strings

We also used a slightly modified version of the procedure described in Section 5.1 to calculate the log-perplexity and empirical entropy of an arbitrary string (that was not generated by the model itself) on the GPT-2 and Llama 3.1 8B models. We first choose a text string that we wish to analyze. We fix an initial fragment of the string as our prompt $\mathbf{X}$. We tokenize the remaining portion of the string excluding the prompt using the model's tokenizer. Let $Y_n$, $n = 1 \ldots N_{max}$ denote the resulting sequence of tokens.

Stating with $n = 1$, we list the probability distribution $p_n(y)$ of the next token $Y_n$ for the model using top-k with $k = 100$ sampling for the $n$'th token. We can then calculate the empirical entropy $h_M(\mathbf{Y}_N)$ and log-deviation $\lambda_M(\mathbf{Y}_N)$ as in Section 5.1. However, to calculate the log-perplexity $l_M(\mathbf{Y}_N)$, we also need the probability that our model will output the actual next token $Y_n$ from the string $\mathbf{Y}$ which may not be included in the list of top-$k$ tokens[4]. When this happens, we replaced the lowest probability entry in the list of top-k tokens with the actual next token from our string along with its probability.

The normalized probability of each of the top-k tokens along with their token-ID and token strings, as well as the probability and ID of the actual next token are saved in a json file. This json file is then processed in exactly the same way as in Section 5.1 to compute $l_M(\mathbf{Y}_N)$, $h_M(\mathbf{Y}_N)$ and $\lambda_M(\mathbf{Y}_N)$ of the sub-strings $\mathbf{Y}_N$, $N = 1 \ldots N_{max}$ for the GPT-2 or Llama 3.1 8B model.

## 5.3 Results and Discussion

Figure 3 shows the log-perplexity and empirical entropy for sub-strings $\mathbf{Y}_N$ for the GPT-2 or Llama 3.1 8B model as a function of $N$ for several text strings generated by the same model. The plots also shows the range $h_M(\mathbf{Y}_N) \pm \lambda_M(\mathbf{Y}_N)$ and $h_M(\mathbf{Y}_N) \pm 2\lambda_N(\mathbf{Y}_N)$. We see that the log-perplexity $l_M(\mathbf{Y}_N)$ stays within the

---

[4]Our choice of top-k sampling is a compromise. The number of tokens in GPT-2 or Llama is quite large ($|\mathcal{T}| \approx 40,000$). If we exhaustively list the full probability distribution of all tokens, the experiment becomes very computationally expensive and slow.

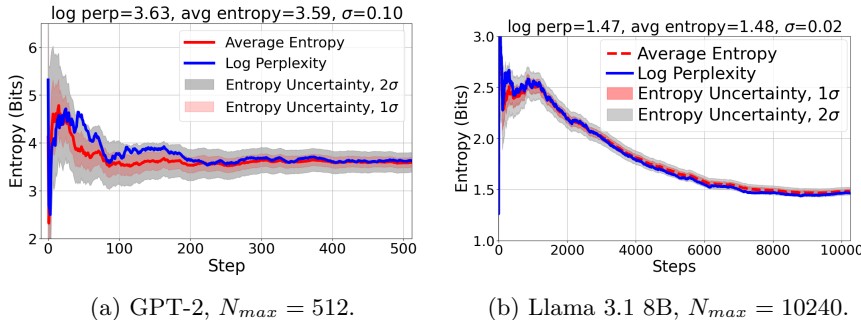

(a) GPT-2, $N_{max} = 512$.      (b) Llama 3.1 8B, $N_{max} = 10240$.

Figure 3: Log-perplexity & empirical entropy for GPT-2 or Llama 3.1, self-generated text, top-100 sampling.

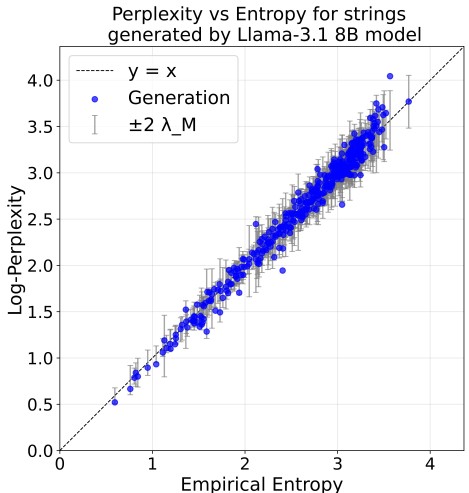

Figure 4: Log-perplexity & empirical entropy of "normal-length" text generated by Llama-3.1 8B.

range $h_M(\mathbf{Y}_N) \pm 2\lambda_M(\mathbf{Y}_N)$ consistently in both cases. The asymptotic convergence is more clearly seen in the long string from the Llama 3.1 8B model[5] in Fig. 3b. Additional plots showing the asymptotic behavior are presented in Appendix B.1. However, we find that very long synthetic texts such as those in Fig. 3b can be ungrammatical and generally of poor quality, as reported elsewhere Holtzman et al. (2020).

While Proposition 3.1 holds regardless of the quality of the text, its practical relevance depends on whether the predicted asymptotic behavior is observable in normal-length texts. Figure 4 shows the log-perplexity $l_M(\mathbf{Y}_N)$ against the empirical entropy $h_M(\mathbf{Y}_N)$ for the Llama-3.1 8B model for several text strings generated by the same model. A total of 450 strings were generated using top-100 sampling from 15 distinct prompts and 30 different random seeds for each prompt. Of these, abnormally short strings of less than 100 tokens were excluded from the analysis. The remaining 383 strings had an average length of 685 tokens and a standard deviation of 321. The clustering of the log-perplexity around the 45° line in Fig. 4 provides direct visual affirmation for Proposition 3.1.

The fact that the perplexity of the generated strings vary over a wide range in Fig. 4 also provides direct evidence for the Assumption 4.1 that the Llama-3.1 8B model is not entropy-maximizing and therefore must have vanishingly small typical sets per Proposition 4.2. Additional evidence for Propositions 4.2 and 4.3 is in Fig. 5 that analyses two long, non-Llama generated strings. The text used in Fig. 5a is an excerpt from a recent article Pelly (2025) in Harper's Magazine, a respected literary publication. The text in Fig. 5b was generated by ChatGPT o1 Pro OpenAI (2024). We see that the log-perplexity $l_M(\mathbf{Y}_N)$ for Llama 3.1 8B is

---

[5]The GPT-2 model cannot generate similarly long texts.

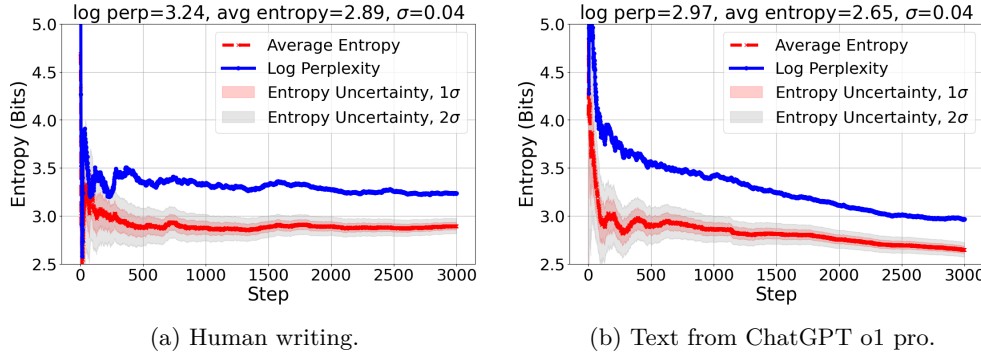

(a) Human writing.

(b) Text from ChatGPT o1 pro.

Figure 5: Log-perplexity and empirical entropy for Llama 3.1 8B of non-Llama-generated strings $\mathbf{Y}$.

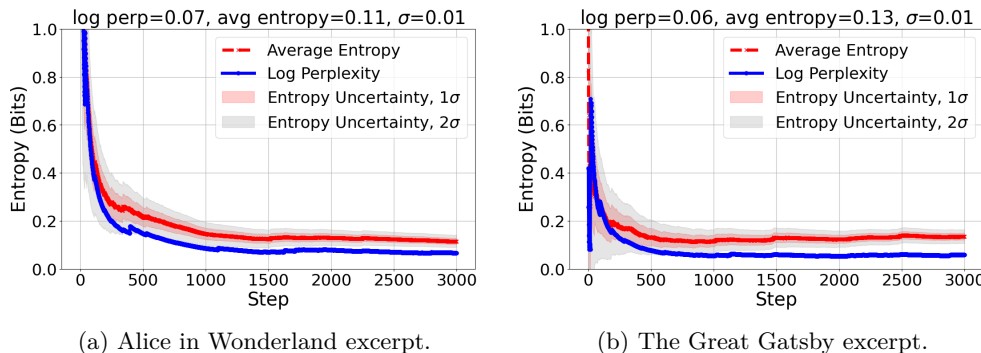

(a) Alice in Wonderland excerpt.

(b) The Great Gatsby excerpt.

Figure 6: Log-perplexity and empirical entropy for Llama 3.1 8B of excerpts from literary classics.

several standard deviations $\lambda_M(\mathbf{Y}_N)$ larger than the empirical entropy $h_M(\mathbf{Y}_N)$ for both strings in Fig. 5 showing that these strings are clearly *under-typical* for the Llama 3.1 8B model. If we assume that the strings in Fig. 5 are randomly sampled from the set of all grammatically correct texts, then this under-typicality is a simple consequence of the smallness of the Llama-3.1 8B model's typical sets.

### 5.4 More Catholic than the Pope

This brings us to the remarkable plots in Fig. 6 which shows the log-perplexity of excerpts from classic texts for Llama 3.1 8B, specifically from Alice in Wonderland and The Great Gatsby in Figs. 6a and 6b respectively. The average entropy for the token distributions in Fig. 6a is an astonishingly small 0.11 bits, which means that the model on average assigns a very high probability ($\sim 99\%$) to the precise sequence of tokens from the excerpt for its predicted next tokens. In other words, *this excerpt has been memorized by the Llama 3.1 8B model* in the sense of eidetic memorization Carlini et al. (2021): with zero temperature, it will output this text verbatim. The same observation also applies to Fig. 6b.

We also see from both plots that the log-perplexity is several standard deviations *below* the empirical entropy making the excerpts slightly, but unambiguously, *over-typical* for the Llama-8B model: the model judges these texts to be more perfect than its own creations! From Proposition 4.3, we can conclude that these excerpts are among the small number of very special strings to be so memorized by the model. Clearly, these texts were part of the training set of the Llama-8B model and they were esteemed highly by its cost function.

In summary, we see from Figs. 3 and 4 that synthetic texts generated by an LLM are *typical* for the model, while most external texts such as in Fig. 5 are *under-typical*. By definition, *over-typical* strings are very special such as the literary classics in Fig. 6 that have been memorized by the model.

# 6 Practical applications: the capybara problem

A simple and direct application of the ideas in this work is for testing whether a given piece of text has "abnormally" high or low perplexity for a language model. Such a test is relevant to many important practical problems. Low perplexity is commonly used as one marker of synthetic text in AI text detection tools (Wu et al., 2025) by testing $l_M(\mathbf{Y}_n) \lessgtr l_{th}$ against a threshold $l_{th}$. However, it turns out that choosing a suitable classification threshold $l_{th}$ is difficult - the so-called "capybara problem" (Hans et al., 2024).

Our theory shows a principled way to do this: with $l_{th} = h_M(\mathbf{Y}_n) + 3\lambda_M(\mathbf{Y}_n)$, Fig. 4 shows that we can accurately distinguish between synthetic text generated by Llama 3.1 8B and non-Llama text such as in Fig. 5 even with relatively short strings of $\approx 500$ tokens. The key insight is to choose *variable* rather than fixed thresholds, i.e. different $l_{th}$ for different strings $\mathbf{Y}_n$.

Our theory can also provide some performance guarantees: e.g. a simple upper-bound on the false negative rate (i.e. failing to detect a Llama-generated string) can be obtained from (17). Note, however, that we cannot provide bounds on *false positive* because the converse to Proposition 3.1 does not hold in general without additional assumptions.

Similar comments apply to the dataset inference problem (Maini et al., 2024): while it is well-known (Carlini et al., 2021) that text strings memorized by an LLM have low perplexity, our theory allows us to rigorously derive threshold values and performance guarantees. Extrapolating from Fig. 6, we posit that *if a long excerpt from a human-written book is over-typical for a language model, it is a near-statistical certainty that the book was part of the training data set for that model.*

# 7 Conclusions

We proposed an asymptotic property that must be satisfied by the perplexity of any long string generated by a language model and provided theoretical and experimental arguments in support of the property. We used this property to extend the information theoretic concept of typical set to language models. The main take-aways are as follows:

1. The classical AEP and the concept of typical sets can be generalized to apply to the outputs of LLMs.

2. These typical sets can be shown to be a vanishingly small subset of possible "good" outputs under very general conditions and this means that LLMs are strongly constrained in what they can generate.

3. Experimentally we see that the asymptotic behavior in the AEP can be observed for LLMs for moderate length texts (a few hundred tokens), and thus may be practically relevant to detecting texts of abnormally high or low perplexity.

4. Synthetic texts from LLMs are random processes with known statistics (unlike natural text), and discovering other laws (e.g. Central Limit Theorems) that apply to LLM outputs is an open research question.

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

# A A Generalized Law of Large Numbers for Random Variables with Arbitrary Dependence

Consider a infinitely long string $\mathbf{Y} = [Y_1 \ Y_2 \ \ldots \ Y_n \ \ldots]$ consisting of a sequence of symbols $Y_n \in \mathcal{Y}$ generated stochastically from a finite dictionary $\mathcal{Y}$. We will denote the substring consisting of the first $N$ symbols as $\mathbf{Y}_N \equiv [Y_1 \ Y_2 \ \ldots Y_N]$ and the joint distribution of the symbols $P(\mathbf{Y}_N)$. We make no assumptions about the distribution $P(\mathbf{Y}_N)$ from which the symbols $Y_n$ are generated. In particular, we do not assume that the symbols are independent or weakly dependent even asymptotically. We also do not assume stationarity i.e. the marginal distributions of symbols $Y_n$ and $Y_m$ can be completely different. We also do not assume any Markovian or conditional independence properties.

## A.1 Conditional Means and Sample Means for Sequences of Dependent Random Variables

Consider a sequence of random variables $\mathbf{Z} = [Z_1 \ Z_2 \ \ldots \ Z_n \ \ldots]$ whose $n$'th element $Z_n$ is a function of the first $n$ symbols $Y_1, Y_2, \ldots, Y_n$ i.e. $Z_n \doteq g_n(\mathbf{Y}_n)$ and $g_n : \mathcal{Y}^n \to \mathbb{R}$ are functions that we will assume to be uniformly bounded i.e. $|g_n(\mathbf{Y}_n)| \leq G < \infty, \ \forall n \in \mathbb{Z}^+, \mathbf{Y}_n \in \mathcal{Y}^n$.

We define the conditional mean and conditional variance of the random variables $Z_n$:

$$\mu_n^{(\mathbf{Y})} \doteq \sum_{y \in \mathcal{Y}} p_n^{(\mathbf{Y})}(y) g_n([\mathbf{Y}_{n-1} \ y]) \equiv E[Z_n | \mathbf{Y}_{n-1}] \tag{28}$$

$$V_n^{(\mathbf{Y})} \doteq \sum_{y \in \mathcal{Y}} p_n^{(\mathbf{Y})}(y) \left( g_n([\mathbf{Y}_{n-1} \ y]) - \mu_n^{(\mathbf{Y})} \right)^2 \equiv E[Z_n^2 | \mathbf{Y}_{n-1}] - \left( \mu_n^{(\mathbf{Y})} \right)^2 \tag{29}$$

where the distribution $p_n^{(\mathbf{Y})}(y) \doteq \Pr(Y_n = y | \mathbf{Y}_{n-1})$. Since the random variables $Z_n$ are discrete-valued and bounded, they have bounded moments for any distribution:

$$|\mu_n^{(\mathbf{Y})}| \leq G, \ V_n^{(\mathbf{Y})} \leq G^2 \tag{30}$$

**A more formal statement.** Let $\mathcal{F}_n$ be the $\sigma$-algebra defined by $\mathbf{Y}_n \equiv [Y_1 \ Y_2 \ \ldots \ Y_n]$ i.e. $\mathcal{F}_n$ is a collection that includes all (mathematically reasonable) events defined on the first $n$ symbols. Then the random variables $Z_n$ as we have defined them are defined by the condition that $Z_m$ is $\mathcal{F}_m$-measurable, and $\mu_m^{(\mathbf{Y})} \equiv \mathrm{E}[Z_m | \mathcal{F}_{m-1}]$.

We need some intermediate results before presenting our main result. First, we define the partial sums of the sequence $Z_n$ and conditional means:

$$S_n \doteq \sum_{m=1}^{n} Z_m, \ \mu_{S_n}^{(\mathbf{Y})} \doteq \sum_{m=1}^{n} \mu_m^{(\mathbf{Y})} \tag{31}$$

Clearly, $\mathrm{E}[Z_n] \equiv \mathrm{E}\left[\mu_n^{(\mathbf{Y})}\right]$ and $\mathrm{E}[S_n] \equiv \mathrm{E}\left[\mu_{S_n}^{(\mathbf{Y})}\right]$. The key idea is that the sequence $Z_m - \mu_m^{(\mathbf{Y})}$ is a martingale difference sequence, and $S_n - \mu_{S_n}^{(\mathbf{Y})}$ is a martingale for any arbitrary distribution of the $Z_m$'s.

**Lemma A.1.** *The variance of the zero-mean random variable $S_n - \mu_{S_n}^{(\mathbf{Y})}$ can be written as sums of contributions from each of the $Z_m$'s. Specifically, we have:*

$$V_{S_n} \doteq E\left[\left(S_n - \mu_{S_n}^{(\mathbf{Y})}\right)^2\right] = \sum_{m=1}^{n} E\left[V_m^{(\mathbf{Y})}\right] \tag{32}$$

*Proof.* We will use the method of induction to prove (32). For $n = 1$, (32) holds trivially. Now, let's assume (32) holds for $n = l$ i.e. assume that:

$$V_{S_l} \equiv \sum_{m=1}^{l} E\left[V_m^{(\mathbf{Y})}\right] \tag{33}$$

Now consider $S_{l+1} \equiv S_l + Z_{l+1}$. Its variance can be written as:

$$V_{S_{l+1}} = E\left[\left(S_l + Z_{l+1} - \mu_{S_l}^{(\mathbf{Y})} - \mu_{l+1}^{(\mathbf{Y})}\right)^2\right] \tag{34}$$

$$= V_{S_l} + E\left[V_{l+1}^{(\mathbf{Y})}\right] + 2\,E\left[\left(S_l - \mu_{S_l}^{(\mathbf{Y})}\right)\left(Z_{l+1} - \mu_{l+1}^{(\mathbf{Y})}\right)\right] \tag{35}$$

We will now show that the last term in (35) is zero. We apply the law of iterated expectations to write:

$$E\left[\left(S_l - \mu_{S_l}^{(\mathbf{Y})}\right)\left(Z_{l+1} - \mu_{l+1}^{(\mathbf{Y})}\right)\right] = \underset{\mathbf{Y}_l}{E}\left[E\left[\left(S_l - \mu_{S_l}^{(\mathbf{Y})}\right)\left(Z_{l+1} - \mu_{l+1}^{(\mathbf{Y})}\right)\Big|\mathbf{Y}_l\right]\right] \tag{36}$$

$$= \underset{\mathbf{Y}_l}{E}\left[\left(S_l - \mu_{S_l}^{(\mathbf{Y})}\right)E\left[\left(Z_{l+1} - \mu_{l+1}^{(\mathbf{Y})}\right)\Big|\mathbf{Y}_l\right]\right] \equiv 0 \tag{37}$$

where we used the fact that $S_l - \mu_{S_l}^{(\mathbf{Y})}$ is a deterministic function of $\mathbf{Y}_l$ and $\mu_{l+1}^{(\mathbf{Y})} \equiv \mathrm{E}\left[Z_{l+1}|\mathbf{Y}_l\right]$.

*Remark.* An alternative way to arrive at (37) is to treat $\mu_{l+1}^{(\mathbf{Y})} \equiv \mathrm{E}\left[Z_{l+1}|\mathbf{Y}_l\right]$ as an estimate of $Z_{l+1}$ given $\mathbf{Y}_l$, and then use the Orthogonality Principle to argue that the estimation error $\left(Z_{l+1} - \mu_{l+1}^{(\mathbf{Y})}\right)$ must be uncorrelated with any function of $\mathbf{Y}_l$. More formally, $\left(S_l - \mu_{S_l}^{(\mathbf{Y})}\right)$ is $\mathcal{F}_l$-measurable and is therefore orthogonal to $(Z_{l+1} - \mathrm{E}\left[Z_{l+1}|\mathcal{F}_l\right])$.

Using (37), we can now simplify (35) to:

$$V_{S_{l+1}} = V_{S_l} + E\left[V_{l+1}^{(\mathbf{Y})}\right] \equiv \sum_{m=1}^{l+1} E\left[V_m^{(\mathbf{Y})}\right] \tag{38}$$

where we used the induction assumption (33). We have shown that if (32) holds for $n = l$, then (32) must also hold for $n = l + 1$, thus completing the induction. □

**Lemma A.2.** *The sequence* $\frac{1}{n}\left(S_n - \mu_{S_n}^{(\mathbf{Y})}\right)$ *converges to zero in probability:*

$$\lim_{n \to \infty} \Pr\left(\frac{1}{n}\left|S_n - \mu_{S_n}^{(\mathbf{Y})}\right| > \epsilon\right) = 0, \ \forall \epsilon > 0 \tag{39}$$

*Proof.* Using Lemma A.1, we can write:

$$V_{S_n} = \sum_{m=1}^{n} E\left[V_m^{(\mathbf{Y})}\right] \le nG^2 \tag{40}$$

using the simple bounds in (30). The variance of $\frac{1}{n}\left(S_n - \mu_{S_n}^{(\mathbf{Y})}\right)$ is simply $\frac{1}{n^2}V_{S_n} \le \frac{G^2}{n} \to 0$ as $n \to \infty$. Equation (39) now follows immediately using the Chebyshev Inequality (Cohen, 2015). □

Lemma A.2 states that the sample means of the two random sequences $\{Z_n\}$ and $\{\mu_n^{(\mathbf{Y})}\}$ must be close to each other asymptotically. Since we have made very few assumptions about the random variables involved, we cannot assume that either of the sequences $\{Z_n\}$ or $\{\mu_n^{(\mathbf{Y})}\}$ themselves converge in any sense to an asymptotic limit. Neither can we assume that their sample means separately converge to a limit.

With an additional assumption stating the existence of one such limit, we can establish a Weak Law of Large Numbers generalized to its most extreme i.e. a statement that the sample mean of a sequence of random variables $\{Z_n\}$ with arbitrary dependence converges asymptotically to a suitably defined average quantity in probability.

**Proposition A.1. A Weak Law of Large Numbers.** *If the average of the conditional means converges to a limit i.e.* $\mu^{(\mathbf{Y})} \equiv \lim\limits_{n\to\infty} \frac{1}{n} \sum\limits_{m=1}^{n} \mu_m^{(\mathbf{Y})}$ *exists, then the sequence* $\frac{1}{n} S_n$ *converges in probability to* $\mu^{(\mathbf{Y})}$:

$$\frac{1}{n} \sum_{m=1}^{n} Z_n \underset{p}{\to} \mu^{(\mathbf{Y})} \quad \text{or} \quad \lim_{n\to\infty} \Pr\left( \left| \frac{1}{n} S_n - \mu^{(\mathbf{Y})} \right| > \epsilon \right) = 0, \ \forall \epsilon > 0 \tag{41}$$

*Proof.* Combining Lemma A.2 with the observation $\frac{1}{n} \mu_{S_n}^{(\mathbf{Y})} \to \mu^{(\mathbf{Y})}$ completes the proof. □

## B  Supplemental Material

### B.1  Additional Experimental Context and Results

In this work, all experiments were ran on a high-performance computing server running Linux (Ubuntu). AMD EPYC 7413 24-Core Processor, 48 logical cores, 96 GB RAM, four NVIDIA A100-SXM4-80GB GPUs. On this setup, each run of a text generation (i.e., running GPT-2 to generate Nmax = 512 tokens, Llama-8B to generate Nmax = 1024 tokens and Nmax = 10240 tokens, their distributions, etc.), file I/O, data analysis and plotting takes several seconds to minutes, with each experiment taking about 90 minutes to run in the case of 10,240 steps. We were able to run a few hundred of these experiments for this paper from which selected examples are presented in Section 5.3.

Figure 7 shows additional results similar to Fig. 3 where we naively varied the seed value of the random number generator from $209 - 214$ for Llama-8B. We present these to show that Fig. 3's results are representative of the general case and were not cherry-picked. In all cases, $K = 100$.

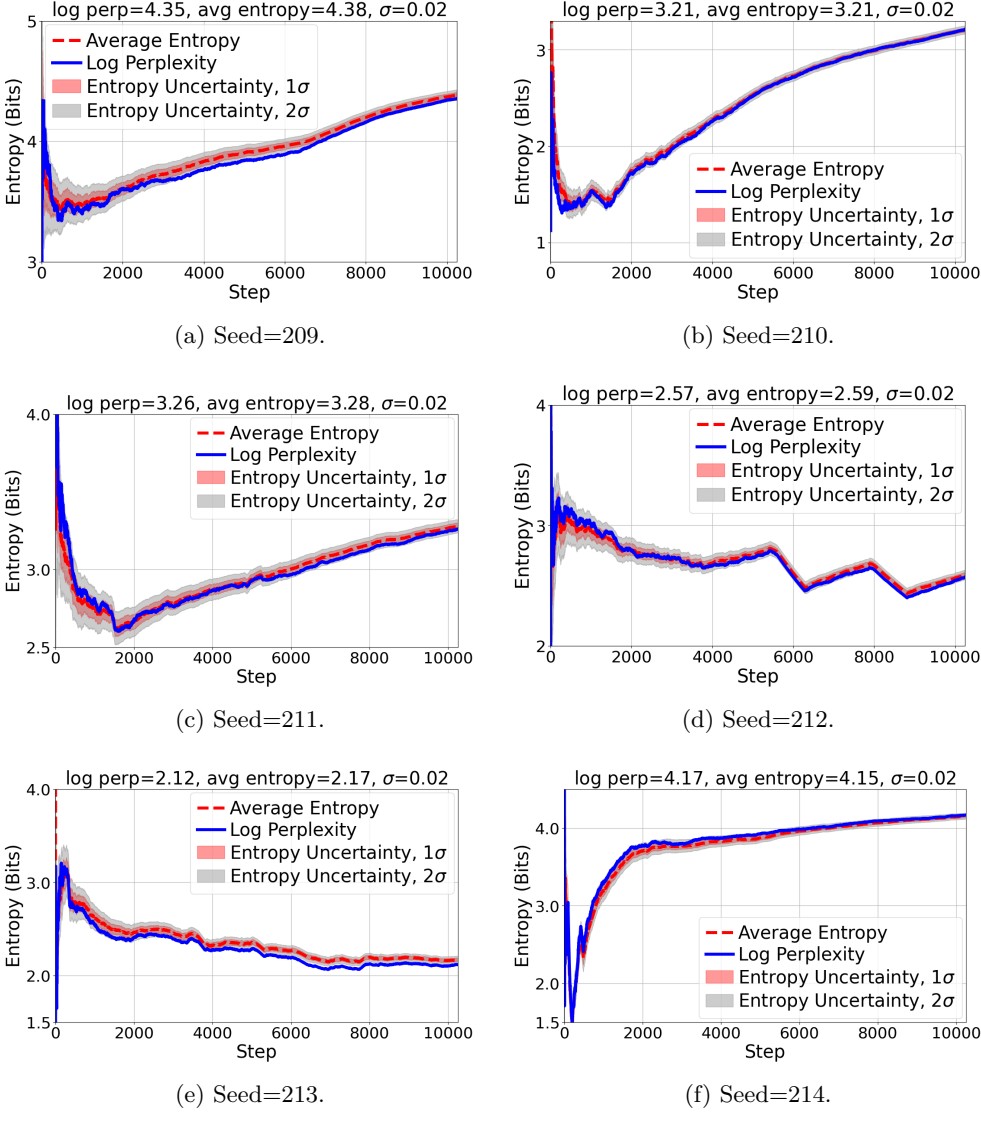

Figure 7: Log-perplexity and empirical entropy for Llama 3.1 of strings generated by the same model with top-100 sampling.

