# OpenReview forum: "AI Stereotypes: An Unequipartition Property for Perplexity in Generative Language Models"
_TMLR — Rejected by TMLR_

### Review · Reviewer_juTt · 2025-05-28

**Summary Of Contributions:**

The submission derives an AEP-style convergence result for sequences generated by large language models, linking the log perplexity (average log probability) of a sequence to the averaged entropy of the model’s output probability distributions across the sequence.  At a high level, this provides a meaningful reference value for the perplexity of a sequence and allows one to define a typical set of sequences as those for which the perplexity and average entropy closely align.

In order to establish the AEP connection, the authors threshold small probabilities so as to bound log probability values, and subsequently apply Chebyshev’s inequality.  In a handful of illustrative examples, the authors show how this can be used to detect whether a string was likely generated by a model and whether a string was memorized by the model during training.

**Audience:**

Yes

**Broader Impact Concerns:**

None.

**Claims And Evidence:**

Yes

**Requested Changes:**

Please see weaknesses above.

**Strengths And Weaknesses:**

## Strengths

- The route to the main result -- Proposition 3.1 -- is marked by clarity and concision.  The authors avoid extraneous theory, and the motivation and exposition are lucid.  This part of the paper is especially strong.

- The bridge from theory to practice is unusually direct, yielding actionable insights about model behavior from a clean information-theoretic foundation.  This paper stands out for bringing theory and application together without the usual compromises.

## Weaknesses

- The experimental evaluation is quite limited.  Results are shown for only a handful of sequences (eight or so), all with identical analysis.  While the Supp claims that hundreds of experiments were run, the results that are included do not reflect the breadth or depth of that effort.

- Section 4 (on the “grammar police”) marks a departure from the paper’s otherwise clear and focused style.  The premise -- to introduce an external grammar-enforcing mechanism -- appears unnecessary, given that a trained model’s output distribution should already encode this information (and more).  Moreover, it’s not clear what value the new construct brings: the theoretical result seems only loosely connected to real model behavior, and the section does not inform any part of the empirical analysis.  I strongly encourage the authors to either clarify the purpose and necessity of this construct or demonstrate its practical value.

---

> ### Author Response · Authors · 2025-06-02
> **We propose to add more experimental data and better motivate the grammar checking**
>
> Dear Reviewer,
>
> Thank you for the review and for your kind comments on the first half of the paper.
>
> To your questions:
>
> > The experimental evaluation is quite limited. Results are shown for only a handful of sequences (eight or so), all with identical analysis. While the Supp claims that hundreds of experiments were run, the results that are included do not reflect the breadth or depth of that effort.
>
> We acknowledge this - we were aiming to maximize high information density and avoid redundancy. We are happy to provide additional data to rectify this. First we should mention that if our paper is accepted, we intend to publish the code for our experiments under an open souce license.
>
> Of the experiments in the paper, Fig. 3 is a direct illustration of Proposition 3.1. The main takeaway from this plot is that we need strings of a few hundred tokens to observe the asymptotic behavior. We can provide additional plots like in Appendix B, but perhaps it'd be more informative if we provide a summary plot that plots the final value of $l_M(Y_n)$ against the final value of $h_M(Y_n)$ for several dozen strings $Y_n$? Please let us know if this seems satisfactory or if you have other suggestions.
>
> The results in Figs 4 and 5 are about _non-typical strings_. These results are related to, but not directly implied by, Propositions 4.2 and 4.3. Scaling these experiments do require going beyond our work in this paper. If there is something specific you'd like to see with these, please let us know.
>
>
> > Section 4 (on the “grammar police”) marks a departure from the paper’s otherwise clear and focused style. The premise -- to introduce an external grammar-enforcing mechanism -- appears unnecessary, given that a trained model’s output distribution should already encode this information (and more). Moreover, it’s not clear what value the new construct brings: the theoretical result seems only loosely connected to real model behavior, and the section does not inform any part of the empirical analysis. I strongly encourage the authors to either clarify the purpose and necessity of this construct or demonstrate its practical value.
>
>
> We recognize that the motivation for Section 4 needs to be more clearly explained. Our goal is to show that the AEP result is not merely a curiosity, that it has significant consequences. Specifically, we want to show that the typical set is vanishingly small just like typical sets defined using the classical AEP for stationary sequences.
>
> Without introducing the concept of a grammar, we can show that the typical set is a vanishingly small subset of the set of all token strings. But we already know that the overwhelming majority of token strings are gibberish, and that even the simplest of modern LLMs e.g. GPT-2 is capable of generating coherent text. Thus this notion of ``smallness'' of the typical set does not seem compelling.
>
> Introducing a grammar constraint allows us to show the stronger result in Proposition 4.2: that the grammatical typical set is a vanishingly small sub-subset (of grammatically correct text) that is already a vanishingly small subset of all token strings.
>
> A good LLM does encode grammar, but we do not require it to be perfect. For a perfect LLM that always generates gramamtical text, we can state simply that the typical set is a vanishing subset of all gramamtical text which itself is a vanishingly small subset of all token strings.
>
> We are happy to clarify or expand on this further here and in the paper.

---

### Review · Reviewer_cUzw · 2025-06-02

**Summary Of Contributions:**

The authors set out to formally study the asymptotic behaviour of strings generated by large language models.

Concretely, the authors prove an asymptotic equipartition property (AEP) type result for discrete stochastic processes using martingale techniques. As the members of the resulting typical sets do not have asymptotically equal probability, the authors call their result an asymptotic un-equipartition property.

Furthermore, the authors also show that their AEP-type result still holds when the sequence of typical sets $T_n$ are slightly restricted: they show that given another sequence of sets $G_n$ satisfying a certain growth condition, the AEP-type result holds for $T_n \cap G_n$ too.

They apply their theoretical results to study large language models and verify that LLMs indeed satisfy the predictions of their results. They also suggest AI-generated text detection as a future practical application of their results.

**Audience:**

Yes

**Claims And Evidence:**

No

**Requested Changes:**

Please address my three major concerns I outline in the weakness section above.

**Strengths And Weaknesses:**

# Strengths

The authors make a much-needed attempt to improve our theoretical understanding of the behaviour of language models with reasonable success. The paper is generally well-written. I particularly liked the authors' writing style.

# Weaknesses

I have three main issues with the paper, all mostly stemming from a lack of precision:
 - I found some of the authors' claims vague or misleading.
 - I found the paper's actual takeaway message unclear.
 - While the writing quality is generally good, the mathematical arguments and notation need improvement.

## The claims

The authors dedicate Section 4 to analyzing stochastic processes restricted to "grammatically correct strings," which they formally identify with decidable languages. However, as far as I can tell, their results do not depend on formal language theory. Rather, the only assumption I could find that was necessary was that the size of the set of length $n$ strings $\mathcal{G}(N)$ with which we are intersecting the typical sets $T_n$ grows as $c 2^n$ for some constant $0 < c = 2^{g_{min}}$.

While I don't think the results the authors derive in Section 4 are uninteresting, if my understanding is correct, then they should rework Section 4 to make it clear that the results don't depend on formal languages. Furthermore, if they wish to keep the connection with grammars, they should discuss which decidable languages satisfy their growth condition. Again, I do not doubt that with the right setup, the authors' results should hold; I'm merely noting their argument does not seem quite tight.

In a similar vein, from Section 2.2 onwards, the authors switch from general probability theoretic language in their propositions to using phrases like "language model" and "tokens." Again, this implied to me that there is some special feature of language models and tokens that are needed for the results to hold. However, in reality, they just replaced "probability distribution" and "random variable" with the more vague terms "language model" and "token", respectively. Instead, the authors should state the propositions with the appropriate rigorous mathematical language and then note that their results apply to contemporary language models in particular.


## The takeaway message
I am not quite sure what the takeaway messages of the paper are. The authors demonstrate that the AEP and related concepts, such as typical sets, can be generalized to a more comprehensive setting using martingale techniques. It is then a fact that their results apply to the probability distributions that LLMs define, but it's not quite clear what LLM-specific insight we gain. Could the authors clarify this, please?

## The arguments and notation

Equations 8 and 9 are wrong/unclear: it's not quite clear what the authors mean by $H(p_A, \alpha) \to H(p_A)$ because $H(p_A)$ still has N-dependence. Furthermore, Eq 9 mixes asymptotic and non-asymptotic statements.

I think a better way of stating the result that the authors want to state is to state the "cross-entropy version" of the AEP, i.e. modify proposition 2.1 to state that for any $A, \beta$, we have $l_B(A) \to H(p_A, \beta)$. Then you can rewrite eq 9 as $\lim l_A(A) = H[\alpha] <= H[\alpha, \beta] = \lim l_B(A)$.

The authors should include a "notations" subsection at the end of Section 1 or the beginning of Section 2, as their work is fairly notation-heavy and uses some non-standard notation, too. Perhaps the most confusing aspect of this was the three different notations the authors use for equality, as they use $=$, $\equiv$ and $\doteq$ without clarifying the difference. Another example is the notation for entropy, where they mix upper and lowercase letter $h$'s and mix probability distributions and random variables as inputs.

Furthermore, it's not quite clear whether the empirical entropy they define in Eq (11) when applied to their setting, e.g., in Eq (12), is meant to be a conditional entropy, i.e., we're only averaging over 1D distributions conditioned on what came before, or if it's meant to be a joint entropy, i.e., we're averaging over strings of length $N$. From the theory part of the paper, it seems that it's meant to be the joint entropy, but then I'm not quite sure how they estimated it in their experiments in Section 5. Could the authors clarify this?

## Minor points
- The authors claim that their techniques do not use any assumptions. However, looking at their proofs in section 3.1 and Appendix A, they do assume uniform boundedness. While this assumption is far more general than the usual statistical assumptions one makes, perhaps the authors should say "minimal" or "common-sense" assumptions rather than "no assumptions".
- Appendix A: "the key idea is that $Z_m - \mu_m$ is a martingale difference sequence" - what definition of martingale difference sequence are the authors using? I thought the usual definition was that the difference sequence comes from a martingale $Y_m$ defined as $Y_m - Y_{m - 1}$, but this is not the structure of $Z_m - \mu_m$.
- Eq (36): the sign in front of the cross-term should be a plus, not a minus because you are expanding $(A + B)^2$, not $(A - B)^2$
- Eq (21): should condition on $Y_n \in \mathcal{G}(n)$, it's not quite clear what $Pr(Y_n \mid \mathcal{G}(n))$ means.
- What is $p_G$ on the extreme right in the inequality below eq 23?
- Small overload of notation: $G$ is used as the grammar-checker-augmented model above eq 20 and as the uniform bound $G$ in the appendix A

## Related work
I think a couple of works that study the related phenomenon of "rule extrapolation" in LLMs should be cited [1, 2].

## Typos
 - $Pr_G$ in equation 24 should just be $Pr$?
 - References are displaying incorrectly (multiple names are shown in lowercase)

## References

1. Reizinger, P., Ujváry, S., Mészáros, A., Kerekes, A., Brendel, W., & Huszár, F. (2024). Position: Understanding LLMs requires more than statistical gegeneralizationarXiv preprint arXiv:2405.01964.

2. Mészáros, A., Ujváry, S., Brendel, W., Reizinger, P., & Huszár, F. (2024). Rule extrapolation in language modeling: A study of compositional gegeneralizationn OOD prompts. Advances in Neural Information Processing Systems, 37, 34870-34899.

---

> ### Author Response · Authors · 2025-06-04
> **We acknowledge all comments and will fix them in a revision; important clarifications below**
>
> Dear Reviewer,
>
> Thank you for the careful reading and detailed review. We appreciate your comment about the writing.
>
>
> > I have three main issues with the paper, all mostly stemming from a lack of precision:
>
> We acknowledge the constructive criticism. Responses to selected comments are below. We will fix the other comments in a revision - thank you for the concrete suggestions and catching several minor issues and typos.
>
>
>
> > The authors dedicate Section 4 to analyzing stochastic processes restricted to "grammatically correct strings," which they formally identify with decidable languages. However, as far as I can tell, their results do not depend on formal language theory. Rather, the only assumption I could find that was necessary was that the size of the set of length $n$ strings $\mathcal{G}(n)$ with which we are intersecting the typical sets $\mathcal{T}_n$ grows as $c^n$ for some constant $c>2^{g\_{min}}$.
>
> This is a perceptive comment that goes to the heart of Section 4. We need to add one detail: that the model generates grammatical texts with a non-zero probability. Formally we need this because otherwise we cannot be (almost) certain that the construction in Fig. 2 terminates in a finite time. Informally, this is to avoid trivial outcomes. Perhaps the best way to see this is with concrete examples.
>
> **Example 1**: The grammar G describes a different language, say Spanish. We would not expect an English language model to generate Spanish text, so the grammatical typical set $\mathcal{T}_n(\epsilon) \cap \mathcal{G}(n)$ will be vanishingly small. Thus Proposition 4.2 holds, but in a very trivial and uninteresting way.
>
> **Example 2** The grammar G describes the Python programming language. Here $\mathcal{G}(n)$ can be precisely specified as valid Python code that ``compiles'' (interprets?) without error. LLMs are capable of generating complex Python code. However, Proposition 4.2 asserts that any model is limited to generating a vanishing fraction of all valid Python programs.
>
> Note that this is much more interesting than saying that the model only generates a small subset of all token strings - the vast majority of tokens strings are gibberish after all.
>
> Your comment (and similar comments from another reviewer) suggests to us that we need to motivate the Section 4 more clearly and we are happy to do so in a revision.
>
>
> > I am not quite sure what the takeaway messages of the paper are. [...] Could the authors clarify this, please?
>
> We can add an explicit list of take-aways:
> 1. The classical AEP and the concept of typical sets can be generalized to apply to the outputs of LLMs.
> 2. These typical sets can be shown to be a vanishingly small subset of possible ''good'' outputs under very general conditions and this means that LLMs are strongly constrained in what they can create.
> 3. Experimentally we see that the asymptotic behavior in the AEP can be observed for LLMs for moderate length texts (a few hundred tokens), and thus may be practically relevant to detecting texts of abnormally high or low perplexity.
> 4. Synthetic texts from LLMs are random processes with known statistics (unlike natural text), and discovering other laws (e.g. Central Limit Theorems) that apply to LLM outputs is an open research question.
>
>
> > I think a better way of stating the result that the authors want to state is to state the "cross-entropy version" of the AEP, i.e. modify proposition 2.1 to state that for any $A, \beta$, we have $l_B(A) \to H(p_A,\beta)$.
>
> This is perfect - it restates our informal statement rigorously with elegance and clarity. Accepted with thanks.
>
>
> > Furthermore, it's not quite clear whether the empirical entropy they define in Eq (11) when applied to their setting, e.g., in Eq (12), is meant to be a conditional entropy, [...]. Could the authors clarify this?
>
> The ``empirical entropy'' $h_M(Y_N)$ defined in Eq (11) depends on $Y_N$ and is therefore a random variable. The term $H(p_n)$ in the RHS of (11) is the entropy of the 1D distribution of the n'th token conditioned on previous tokens. (This is *not* the same as conditional entropy, a term we want to be careful not to overload.) Eq (12) is indeed the joint entropy (a deterministic number) which is defined as an average over strings of length $N$.
>
> There is some unavoidable complexity here, but we accept the burden of explaining things clearly and will do our best to do so in a revision.

---

> > ### Comment · Reviewer_cUzw · 2025-06-10
> >
> > Thank you for addressing some of my concerns so far.
> >
> > Unfortunately, I am not yet convinced by your reply to my comment about Section 4. My point was that there is a technical gap in the paper: my understanding is that your results hold when intersecting the sequence of typical sets with any sequence of sets whose size grows as $\Omega(e^n)$, and you didn't refute this in your response. On the other hand, you base your high-level argument on decidable languages. However, there is no technical result connecting the two:
> >  - Are the sequences of sets with your exponential growth condition equivalent to the set of decidable languages? (I doubt it)
> >  - As a slightly weaker but still interesting question: Are all sequences of sets with your required exponential growth condition equivalent to a subset of decidable languages? (I doubt this, too, but I am more hopeful)
> >  - Under what conditions do decidable languages have strings of length $n$ grow as you need?
> >
> > As I suggested in my review, I would avoid presenting the results so tightly linked with decidable languages unless the proofs truly rely on them. This is because, in its current form, I find the results misleading, and you could also state a potentially more powerful result instead.
> >
> > This is not considering the age-old philosophical troubles that your Example 1 raises about natural languages being decidable. I find your Example 2 more interesting, though I am still concerned about its usefulness: it's not entirely clear whether this is a limitation or not. For example, I wouldn't expect an LLM to output the codebase for a webshop followed by the AI that controls a Mars rover, despite the possibility of forming these in valid Python (or as a combination of programming languages).
> >
> > Regarding the empirical entropy, how did you estimate the joint entropy in Figure 3?

---

> > > ### Author Response · Authors · 2025-06-11
> > > **formal grammar not needed for math result, but may be useful for intuition and interpretation**
> > >
> > > Thank you for the follow-up.
> > >
> > >
> > > > Unfortunately, I am not yet convinced by your reply to my comment about Section 4. My point was that there is a technical gap in the paper: my understanding is that your results hold when intersecting the sequence of typical sets with any sequence of sets whose size grows as
> > > $\Omega(e^n)$ and you didn't refute this in your response.
> > >
> > > First we want to clarify that we agree with your observation above: the main result Proposition 4.2 applies to arbitrary large sets $\mathcal{G}(n)$ and is not limited to decidable languages. We also agree that we do not need any of the properties of formal languages. We do not seek to refute any of this and we are happy to restate the proposition 4.2 in the more general form.
> > >
> > > The question is whether we still need the construction in Fig. 2 (which assumes a decidable grammar) to make interesting practical statements such as ''the model can only generate a small fraction of all possible sonnets''.
> > >
> > > We introduced Fig. 2 with a very specific goal: to show that typical sets are small in a *meaningful* way, specifically, there is a large class of ''good'' outputs (not just gibberish token strings) that the model will not generate. The ''good'' strings are defined by the set $\mathcal{G}(n)$. We want to say that most ''good'' strings in $\mathcal{G}(n)$ are not in $\mathcal{T}_n(\epsilon)$ and therefore will almost never be generated.
> > >
> > > Fig. 2 allows us to do this for decidable grammars in an explicitly constructive way: the augmented model G will almost certainly generate strings only from $\mathcal{T}_n(\epsilon) \cap \mathcal{G}(n)$. The question is how do we do this for general large sets $\mathcal{G}(n)$ without the analytical device of the augmented model G. The best we can do is to try and show that $\Pr(\mathcal{T}_n(\epsilon) \cap \mathcal{G}(n)| Y_n \in \mathcal{G}(n)) \to 1$ without assigning an operational meaning to this probability.
> > >
> > > Ultimately we think this is an aesthetic question. Is it satisfactory if we state the math results in the general form, but still retain the construction in Fig. 2 to interpret the results intuitively?
> > >
> > >
> > > > Are all sequences of sets with your required exponential growth condition equivalent to a subset of decidable languages? (I doubt this, too, but I am more hopeful)
> > >
> > > We agree. We can imagine a sequence $\mathcal{G}(n)$ where for each $n$ we choose at random say $2^n$ token strings. Such a sequence is not decidable by any finite length program for a Turing machine. So you are correct to point out that the assumption of decidability is restrictive in a non-trivial way.
> > >
> > >
> > > > Regarding the empirical entropy, how did you estimate the joint entropy in Figure 3?
> > >
> > > It is calculated from the definition in (11) where the $p_n$'s are obtained following the procedure described in Section 5.1.
> > >
> > > We appreciate the discussion and we are happy to elaborate further.

---

### Review · Reviewer_vA18 · 2025-06-20

**Summary Of Contributions:**

The focus of the paper is on the concentration properties exhibited by the empirical entropy of strings sampled from a language model. The paper proposes a variation of the asymptotic equipartition property (AEP) to account for the fact that the tokens in samples from a language model are not i.i.d., and attempts to prove that language models exhibit this "unequipartition" property for sufficiently long strings. The paper also contributes experiments 1) estimating the distribution of the observed entropy of strings sampled from various large language models; 2) estimating the perplexity of certain texts (e.g., Alice in Wonderland and The Great Gatsby) under various large language models; and 3) showing that text generated by Llama 3 8B has lower entropy than text not generated by Llama 3 8B.

**Audience:**

Yes

**Broader Impact Concerns:**

I have no concerns regarding the ethical implications of the work.

**Claims And Evidence:**

No

**Requested Changes:**

Given TMLR's emphasis on claims and evidence, my main concerns revolve around 1) the validity of the theoretical result and 2) the paper's interpretation/analysis of the experimental results. I am open to changing my recommendation to accept if the authors are able to address the concerns I have outlined above ("Weaknesses").

**Strengths And Weaknesses:**

Strengths:
The paper is thoughtfully written: the writing is clear and the authors make an attempt to guide readers through key concepts by building up to the main results through simple toy cases and informal analyses.

Weaknesses:
There are a few major weaknesses.

First, at various points the paper makes sweeping and apparently unsubstantiated claims. For example, the paper states:
> Natural languages are shaped by complex social, psychological and biological processes and they can only ever be approximated by computational models. In contrast, LLMs are machines whose internals, while complex, are in theory completely knowable.

Why is it the case that these processes "can only ever be approximated by computational models"? What is the definition of a computational model? Why does it matter that we can observe the "internals" (activations/weights?) of an LLM? (The paper does not appear to mention LLM internals elsewhere, e.g., in any of the main theorems or empirical results)

Another example:
> Our theory can also provide performance guarantees: e.g. a simple lower-bound on the false negative rate (i.e. failing to detect a Llama-generated string) can be obtained from (18)

It would be helpful to precisely state these guarantees. Otherwise, it is not clear what/if anything the theory does guarantee regarding control over false positives for detecting Llama-generated strings. In particular, it appears that in order to compute (18) we would need to know the prompt that produced the string, but in practice this prompt will typically not be available (i.e., if a student prompts a language model to do their homework for them, they will likely submit the homework but not the prompt). Text detection appears to be one of the main applications the authors highlight to argue for the significance of their theoretical results, so it would be good to flesh out these implications more precisely.

Second, the related work is a bit too broadly scoped. Instead of having a subsection on "Natural Language Processing" as a whole, it would be more useful (and provide better context regarding the significance of the contributions of the paper) to discuss the most relevant related work (e.g., 1-2 most closely related papers) in depth.

Third, there appear to be some basic issues with the problem formulation. In particular, the paper defines the token sequence Y as a deterministic function of a prompt X and pseudorandom sequence W, but then immediately afterwards treats Y as a random variable associated with a probability distribution. Since W is pseudorandom, everything should be deterministic. (The simple solution would be to just do away with W entirely, since it seems irrelevant to the main results, and treat Y as a proper random variable.) At face value, it appears that all of the main theoretical results in the paper (i.e., the concentration results) are invalid since Y does not have a distribution.

Finally, it is not obvious that the experiments support the authors main claims that entropy concentrates as sequence length grows large. In particular, if this were the case, then should we not expect the confidence bands in Figure 3 to narrow with sequence length? (It appears there is some narrowing, but eg in 3a the size of the band seems to stabilize after ~250 steps.) Furthermore, Section 5.4 seems to contradict the paper's claim elsewhere that one can detect text generated by a language model based on whether it has lower perplexity than other text. Indeed, others (https://arxiv.org/abs/1906.05664) have found that the entropy/perplexity of language models' own generations often exceeds that of text in the training data. The paper does mention:
> if a long excerpt from a human-written book is over-typical for a language model, it is a near-statistical certainty that the book was part of the training data set for that model.
But it would be worth more systematically investigating the tension between the existence of over-typical sequences and detecting synthetic text.

---

> ### Author Response · Authors · 2025-06-23
> **Clarification on the two main questions about interpretation of experimental results using the theory**
>
> Dear Reviewer,
>
> Thank you for the review.We appreciate the positive comments about the writing. Below are detailed responses to your comments. We will first address your two main concerns about (a) the validity of the theoretical result and (b) interpretation of experiments. We will address your other concerns in a separate comment.
>
>
>
>
> > Finally, it is not obvious that the experiments support the authors main claims that entropy concentrates as sequence length grows large. In particular, if this were the case, then should we not expect the confidence bands in Figure 3 to narrow with sequence length? (It appears there is some narrowing, but eg in 3a the size of the band seems to stabilize after ~250 steps.)
>
>
> You are correct: we do expect the confidence bands to narrow with length. To confirm this we would need to continue generating a longer string. But the simple GPT-2 model used in Fig. 3a is not able to generate longer strings. The more capable Llama model is able to generate longer strings and we can see the predicted behavior in Fig. 3c and 3d. More such plots are shown in the Appendix B.
>
> Overall we take from your comment (and similar comments from another Reviewer) that the paper will benefit from a richer set of experimental results. We will generate and share such results soon for a revision.
>
>
>
> > Furthermore, Section 5.4 seems to contradict the paper's claim elsewhere that one can detect text generated by a language model based on whether it has lower perplexity than other text. Indeed, others (https://arxiv.org/abs/1906.05664) have found that the entropy/perplexity of language models' own generations often exceeds that of text in the training data.
>
> Rather than a dichotomy between AI text and natural text, we need a _trichotomy_ between AI text, natural text and training text. Fig, 3 shows AI text, Fig. 4 natural text and Fig. 5 training text.
>
> The convergence in Fig. 3 is mandated by Proposition 3.1 - lack of convergence here would directly contradict our theory and would point to a math error or invalid assumption.
>
> Our theory does not directly predict the results in the Fig. 4 and 5 because we have not considered the converse to Proposition 3.1. However, if we add a separate assumption that the natural texts in Fig. 4 are randomly chosen from the set of all natural texts, then the observed results are explained by Proposition 4.2.
>
> But texts from the training dataset are not ``random samples'' and the above reasoning does not apply to those. Empirically, Fig. 5 shows that come classic texts have been memorized by the Llama model. Proposition 4.3 states that there cannot be too many such memorized texts.
>
> We are happy to clarify further here and in the paper.

---

> ### Author Response · Authors · 2025-06-23
> **Clarifications on other review comments**
>
> Below are responses to your remaining concerns.
>
> > First, at various points the paper makes sweeping and apparently unsubstantiated claims. For example, the paper states:
> > > Natural languages are shaped by complex social, psychological and biological processes and they can only ever be approximated by computational models.
>
> > Why is it the case that these processes "can only ever be approximated by computational models"? What is the definition of a computational model?
>
> This comment is intended simply to emphasize our exclusive interest in synthetic text and make a contrast with the (much harder) problem of analyzing natural text.
>
> ''Computational models'' refers to the class of models reviewed in the preceding section (Sec 1.2) e.g. ngrams, HMMs, modern LLMs and also structural models based on formal grammars. We do not believe that this claim is sweeping - such limitations of models representing natural languages have been acknowledged since the early days of modern linguistics. See e.g. the critiques of FAHQMT (Fully Automatic High Quality Machine Translation) in _Bar-Hillel, Y. (1960). "A Demonstration of the Non-feasibility of Fully Automatic High Quality Translation."_.
>
>
>
> > > In contrast, LLMs are machines whose internals, while complex, are in theory completely knowable.
>
> > Why does it matter that we can observe the "internals" (activations/weights?) of an LLM? (The paper does not appear to mention LLM internals elsewhere, e.g., in any of the main theorems or empirical results)
>
> The ''internals'' used in our work are the activations of the softmax layer from which the token distributions $p_n(y)$ are obtained. This is all we need for our results in this work, but with full knowledge of model weights, activations etc, we expect we can make many more interesting observations about model behavior.
>
>
>
> > It would be helpful to precisely state these guarantees. Otherwise, it is not clear what/if anything the theory does guarantee regarding control over false positives for detecting Llama-generated strings
>
> As stated in Sec 6, Eq (18) guarantees a _false negative_ rate less than $\frac{1}{9} \approx 11$ % for a threshold $l_{th} = h_M(Y_n) + 3 \lambda_M(Y_n)$. Our theory by itself cannot provide bounds on false positives (since we don't consider a converse to Prop 3.1) and will need to be supplemented by additional assumptions. Note also that (18) is a very weak bound. These are open problems for future work.
>
>
>
> > In particular, it appears that in order to compute (18) we would need to know the prompt that produced the string, but in practice this prompt will typically not be available
>
> Excellent question! As explained in Sec 5.2, we get around this in our own experiments with a simple expedient: we just use an initial fragment of the text as our prompt.
>
>
>
> >  (i.e., if a student prompts a language model to do their homework for them, they will likely submit the homework but not the prompt)
>
> We agree. More broadly, we are under no illusions about the many challenges involved in translating a theoretical work like ours into a practical method for detecting AI text. This is another open problem for future works and we make no claims to have working solutions to such real-world problems.
>
>
> > Second, the related work is a bit too broadly scoped. Instead of having a subsection on "Natural Language Processing" as a whole, it would be more useful (and provide better context regarding the significance of the contributions of the paper) to discuss the most relevant related work (e.g., 1-2 most closely related papers) in depth.
>
> Thank you for this suggestion. In light of your comments above, we could discuss the works of John Pierce and Bar-Hillel on computational models of language?
>
>
>
> > Third, there appear to be some basic issues with the problem formulation. In particular, the paper defines the token sequence Y as a deterministic function of a prompt X and pseudorandom sequence W, but then immediately afterwards treats Y as a random variable associated with a probability distribution.
>
>
> The pseudo-randomness is relevant for one very narrow reason - it allows us to reproduce random experiments exactly by controlling the PRNG seed. Since pseudorandom variables satisfy the same statistical laws as random variables, we simply treat them as rvs for simplicity of terminology.
>
> However, we take your point: this is not essential to the paper and we are happy to remove it for clarity.

---

### Author Response · Authors · 2025-06-30
**Revision submitted; feedback from Reviewers requested**

Dear Reviewers,

We have now prepared and uploaded a draft revision of our manuscript in response to the your reviews and comments.

We wanted to share this quickly to facilitate discussion. We plan to do a careful proof-read later, but we kindly request your comments on these changes and whether they address your concerns.

Below is a summary of changes.


 - To Reviewer juTt: Please see Fig. 4 and related discussion for a much richer set of experimental results. We have also attempted to motivate Sec 4 better. Specifically in Sec 4.3, we apply our theory to three examples of grammars: (a) the Python programming language, (b) mathematical proofs in First Order Logic and (c) a form of lyrical poetry.


 - To Reviewer cUzw: We have fixed typos and cited suggested references. We have also fixed the capitalization issues in the bibliography. Prop 2.1 is rephrased as suggested. Sec 1.4 was added to clarify notation. An explicit list of takeaway messages was added to the Conclusion. Most importantly, Sec 4 was reorganized to state results without using formal grammars until they are needed.


 - To Reviewer vA18: We have rewritten the lit survey in Sec 1.2 to be more focused. We have removed references to pseudo-randomness for clarity. We have added new experimental results (see Fig. 4 and related discussion). We added a paragraph to Sec 5 summarizing synthetic vs natural vs training text, and clarified the scope of performance guarantees that our theory can provide in Sec 6.

---

### Decision · Action_Editor_yJcT · 2025-08-12

**Recommendation:** Reject

**Additional Comments:**

For a major revision, I would expect at least the following changes:
- Sweeping claims such as the following cannot be substantiated with two citations to papers from the 1960s. I would expect these sorts of grand statements to either be omitted, clarified, or more precisely stated and substantiated.
> Natural languages are shaped by complex social, psychological and biological processes and they can only ever be approximated by computational models.
- Claims of performance guarantees should be precisely specified, unless they would be clearly evident to a wide audience. One reviewer felt the following claim was not clearly justified, and this was not addressed in the revision. Such claims should be removed or expanded upon in a major revision.
> Our theory can also provide some performance guarantees: e.g. a simple upper-bound on the false negative
rate
- Make sure that key concepts are clearly defined as they are introduced. For example, "Un-Equipartition Property" is introduced without definition in the introduction.
- Avoid informal definitions. For example, the dictionary $\mathcal{G}(n)$ is introduced as a set of "grammatical" strings. Grammatical should not be defined in quotations, and should be defined precisely.

Making these changes wouldn't guarantee a recommendation of acceptance, but it would help make the paper more accessible and impactful. All of the reviewers felt that this paper had promise. So, I urge the authors to review the recommendations carefully and prepare a new manuscript that is more narrowly and precisely pitched.

**Audience:**

Yes

**Audience Explanation:**

This paper investigates the concentration properties of a language model's perplexities. The result that these must approach the average entropy of its tokens is an interesting result that may be of interest to practitioners and theorists.

**Claims And Evidence:**

No

**Claims Explanation:**

In their initial reviews, all reviewers were concerned about the clarity and accuracy of the writing. In their final recommendations, reviewers were divided. Some felt that the revised manuscript and rebuttal had addressed their concerns. Others felt that the writing still made a number of implicit, unsupported claims.

I ultimately believe that the concerns have not been fully addressed in the current revision, and am recommending rejection with the possibility of major revision.

**Resubmission Of Major Revision:**

The authors may consider submitting a major revision at a later time.